# Mycobacterial HelD connects RNA polymerase recycling with transcription initiation

Tomáš Kovaľ [1,5], Nabajyoti Borah[2,3,5], Petra Sudzinová [2], Barbora Brezovská[2], Hana Šanderová [2], Viola Vaňková Hausnerová[2,3], Alena Křenková[4], Martin Hubálek[4], Mária Trundová[1], Kristýna Adámková[1], Jarmila Dušková[1], Marek Schwarz [2], Jana Wiedermannová [2], Jan Dohnálek[1] ✉, Libor Krásný [2] ✉ & Tomáš Kouba [4] ✉

Mycobacterial HelD is a transcription factor that recycles stalled RNAP by dissociating it from nucleic acids and, if present, from the antibiotic rifampicin. The rescued RNAP, however, must disengage from HelD to participate in subsequent rounds of transcription. The mechanism of release is unknown. We show that HelD from *Mycobacterium smegmatis* forms a complex with RNAP associated with the primary sigma factor $\sigma^A$ and transcription factor RbpA but not CarD. We solve several structures of RNAP-$\sigma^A$-RbpA-HelD without and with promoter DNA. These snapshots capture HelD during transcription initiation, describing mechanistic aspects of HelD release from RNAP and its protective effect against rifampicin. Biochemical evidence supports these findings, defines the role of ATP binding and hydrolysis by HelD in the process, and confirms the rifampicin-protective effect of HelD. Collectively, these results show that when HelD is present during transcription initiation, the process is protected from rifampicin until the last possible moment.

Transcription is the first step of gene expression where information stored in DNA is transcribed into RNA by RNA polymerase (RNAP). RNAP in bacteria consists of several core subunits ($\alpha_2\beta\beta'\omega$) and a $\sigma$ factor[1]. The RNAP core possesses catalytic activity and $\sigma$ provides it with specificity for promoter DNA that is essential for transcription initiation[2]. Topologically, RNAP contains three channels: (i) the primary channel that consists of several parts of which most pertinent for this study are the β'-clamp mobile feature and the region where downstream DNA (dwDNA) and the DNA–RNA hybrid bind; (ii) the secondary channel through which nucleoside triphosphates (NTPs) enter the active site (AS); and (iii) the RNA exit channel[3]. Functioning of RNAP is then regulated by its interactions with DNA, by small molecule

effectors (e.g., ppGpp, initiating NTPs [iNTPs]), and various transcription factors (small RNAs, proteins) that bind to/interact with various regions of RNAP[4–7].

HelD is a protein transcription factor that binds and hydrolyzes ATP/GTP by a conserved NTPase Rossmann fold 1A–2A heterodimer[8]. HelD associates with RNAP by penetrating the primary channel with its β'-clamp opening domain (CO-domain) and the secondary channel with its N-terminal (N-term) domain. The NTPase 1A protomer physically connects the $HelD_{N-term}$ and CO-domain on the periphery of both the primary and secondary channel and together with the 2A protomer configures the mutual orientation of $HelD_{N-term}$ and CO-domain. The HelD–RNAP interaction within the primary channel is incompatible

[1]Institute of Biotechnology of the Czech Academy of Sciences, Průmyslová 595, 252 50 Vestec, Czech Republic. [2]Institute of Microbiology of the Czech Academy of Sciences, Vídeňská 1083, 142 20 Prague, Czech Republic. [3]Department of Genetics and Microbiology, Faculty of Science, Charles University, Viničná 5, 128 44 Prague, Czech Republic. [4]Institute of Organic Chemistry and Biochemistry of the Czech Academy of Sciences, Flemingovo náměstí 542/2, 160 00 Prague, Czech Republic. [5]These authors contributed equally: Tomáš Kovaľ, Nabajyoti Borah. ✉e-mail: Jan.Dohnalek@ibt.cas.cz; krasny@biomed.cas.cz; tomas.kouba@uochb.cas.cz

with the presence of nucleic acids[9]. Indeed, HelD was shown to remove stalled RNAPs from DNA, thereby recycling RNAP and the template for subsequent rounds of transcription[10]. Such stalled complexes can arise due to obstacles on DNA or to RNAP inhibition by rifampicin-like antibiotics. Rifampicin binds to a pocket where the DNA–RNA hybrid binds and during initiation of transcription prevents the nascent RNA from elongating beyond 2–3 nucleotides (nt)[11]. In both cases, stalled RNAPs block transcription and pose a threat to genome stability due to collisions with the replication machinery.

Currently, three classes of HelD proteins are recognized[12]. Relevant for this study are class II HelD proteins (HelR is a recently proposed alternative name) that are found in industrially and medicinally important Actinobacteria. This class is characterized by the presence of a topological feature, the primary channel loop (PCh-loop), which reaches to the AS of RNAP where a two nt long duplex of the nascent DNA–RNA hybrid would be positioned[9]. The presence of HelD in this area not only interferes with the AS itself and the DNA–RNA hybrid binding but also induces displacement of rifampicin from its binding pocket[13,14]. In this way, HelD releases the rifampicin-stalled transcription initiation complexes[15] and functions as a target protection mechanism of antibiotic resistance[16].

The available structures of HelD with RNAP reveal extensive binding interfaces occluding critical functional parts of RNAP and resulting in tight HelD–RNAP complexes[9,17,18]. The exact mechanism of how RNAP is released from the grip of HelD so that it can participate in the next round of transcription is currently unknown[15].

Here, to comprehensively address this question in the context of transcription initiation of mycobacterial RNAP, we first performed an unbiased screen in *Mycobacterium smegmatis (Msm)* for complexes containing HelD. This screen confirmed previous in vitro results that HelD is in complexes with (i) RNAP, (ii) σ[A], the primary σ factor, and (iii) RbpA[9], a transcription activator of the RNAP–σ[A] holoenzyme that is also involved in antibiotic resistance[19,20]. Moreover, the experiments revealed that HelD can also be in complexes with σ[B], an alternative σ factor[21]. Finally, it showed that the presence of HelD on RNAP excludes the presence of CarD, an essential global regulator that activates RNAP by affecting the open-promoter complex[22,23]. We next verified the coexistence of RNAP–σ[A]–RbpA–HelD structurally by cryogenic electron microscopy (cryo-EM). Subsequently, by a series of cryo-EM snapshots, we visualized the sequence of events leading to transcription initiation where the RNAP–σ[A]–RbpA–HelD complex first binds promoter DNA outside the primary channel, and subsequently loads DNA into it. Concomitantly, the respective HelD domains are released from the primary channel while the presence of the HelD$_{N-term}$ secondary channel-specific domain is compatible with a DNA-loading intermediate. Finally, the completion of DNA loading into RNAP fully displaces the entire HelD protein and liberates RNAP for transcription initiation. Biochemical experiments then demonstrated that the HelD release is stimulated by ATP binding; ATP hydrolysis further facilitates this process. Additionally, the promoter DNA itself contributes to expulsion of HelD. Finally, the effect of HelD on transcription in vitro in the presence/absence of rifampicin was characterized.

Taken together, this study connects termination of transcriptionally incompetent complexes with transcription initiation. It mechanistically explains the involvement of mycobacterial class II HelD in initiation of transcription, and the stepwise process of its disengagement from RNAP. During this process, the presence of HelD on RNAP helps protect RNAP against rifampicin.

## Results

### Search for interaction partners of HelD
To identify direct and indirect interaction partners of HelD in vivo, we used an *Msm* HelD-FLAG strain and its parent strain without any tag, and performed pull-down experiments from cells in exponential and stationary phase of growth, followed by identification of the proteins by mass spectrometry (Fig. 1a, b; Supplementary Tables 1 and 2). The results identified subunits of RNAP including σ[A] and σ[B], and transcription factor RbpA. Notably, we did not detect the other mycobacterial transcription initiation-specific factor, CarD (Fig. 1b, c). To provide more insight into the binding of HelD and σ[A] to RNAP, we performed additional experiments and determined the relative levels of HelD and σ[A] on the RNAP core in exponential phase and from three time points in stationary phase. These levels remained more or less constant, documenting that the complex is present in the cell under different physiological conditions (Supplementary Fig. 1).

We concluded that HelD was in complexes with the RNAP core, the primary sigma factor, σ[A], and the transcription factor RbpA, confirming our previous in vitro results[9]. Additionally, HelD can be in complexes also with σ[B]. This is an alternative σ factor that is active both in exponential and stationary phases and directs transcription of stress-related as well as some housekeeping genes[24,25]. Both σ[A] and σ[B] contain conserved domains σ$_2$-σ$_3$-σ$_4$[26]. The main topological difference between σ[A] and σ[B] is in the presence of the σ[A]-specific N-terminal domain[27]. This domain differs substantially from analogous domains in primary σ factors in other species (e.g., *E. coli* and *B. subtilis*), where it is structured and termed domain 1.1[28,29]. The mycobacterial N-terminal domain is mostly unstructured with only one helix detectable at its C-terminus (σ[A]$_{N-helix}$)[30]. Finally, the presence of HelD and CarD on RNAP seems to be mutually exclusive. Hence, to provide insights into functioning of HelD during transcription initiation as well as into the mode of its release from RNAP, we selected for subsequent studies the quaternary RNAP–σ[A]–RbpA–HelD complex.

### HelD and σ[A] specifically interact in the primary channel of RNAP
In order to structurally visualize the quaternary complex, we reconstituted the RNAP core with excess of HelD, σ[A], and RbpA, subsequently purified the complex by size exclusion chromatography (SEC; Supplementary Fig. 2) and analyzed it by cryo-EM (Supplementary Figs. 3 and 4). Two major 3D classes of interest were identified. The first class (hereafter HelD–holo-I), at an overall resolution 3.11 Å, visualized complete HelD bound to RNAP together with σ[A]$_{N-helix}$, σ[A]$_2$, σ[A]$_3$, and both RbpA N- and C-terminal domains (Fig. 1d, e, Supplementary Movie 1). HelD is engaged in both the primary and secondary channel, and particularly the HelD CO-domain is wedged in-between the β-lobe and β′-clamp in a very similar way as in the previously identified State I HelD–RNAP complex[9]. The second class (hereafter HelD–holo-II), at an overall resolution 3.14 Å, visualized complete HelD bound to RNAP together with σ[A]$_{N-helix}$, but only σ[A]$_2$ and RbpA C-terminal domains (Fig. 1f, Supplementary Movie 1). The HelD configuration in this class resembles the previously identified State II HelD–RNAP complex where, in addition to the HelD CO-domain wedged into the primary channel, the HelD PCh-loop interferes with the RNAP AS[9].

In contrast to the previously published HelD–RNAP core complexes[9], the current two structures reveal how HelD and σ[A] specifically interact in the context of the RNAP core. In both classes, σ[A]$_2$ interacts with the canonical binding site on the β′-clamp coiled-coil domain. In the previous HelD–RNAP structures, the CO-domain of HelD pointed towards the σ[A]$_2$ binding site but did not make any specific interactions with the coiled-coil domain, which was accompanied by a low local resolution of this region. The presented structures show that the σ[A]$_2$ domain, when bound to β′-clamp, constitutes a specific interaction platform for the CO-domain tip (Fig. 1g). Together, HelD and σ[A] share ~1300 Å$^2$ of buried surface area. In detail, the HelD CO-domain helix-turn-helix (HTH) tip is wedged between the N-terminus of σ[A] starting at σ[A]/Phe140, the σ[A]$_{N-helix}$ and the σ[A]$_2$ helical bundle (Fig. 1h). The hydrophobic interactions of HelD/Tyr347 with σ[A]/Phe140 and σ[A]/Trp142 define the C-terminal border of the observed σ[A] N-terminus. The following σ[A]$_{N-helix}$ lies across the CO-domain helix-turn-helix, which is buttressed from its side by helices 1 and 4 of the σ[A]$_2$ helical bundle. In contrast to the previously observed HelD–RNAP

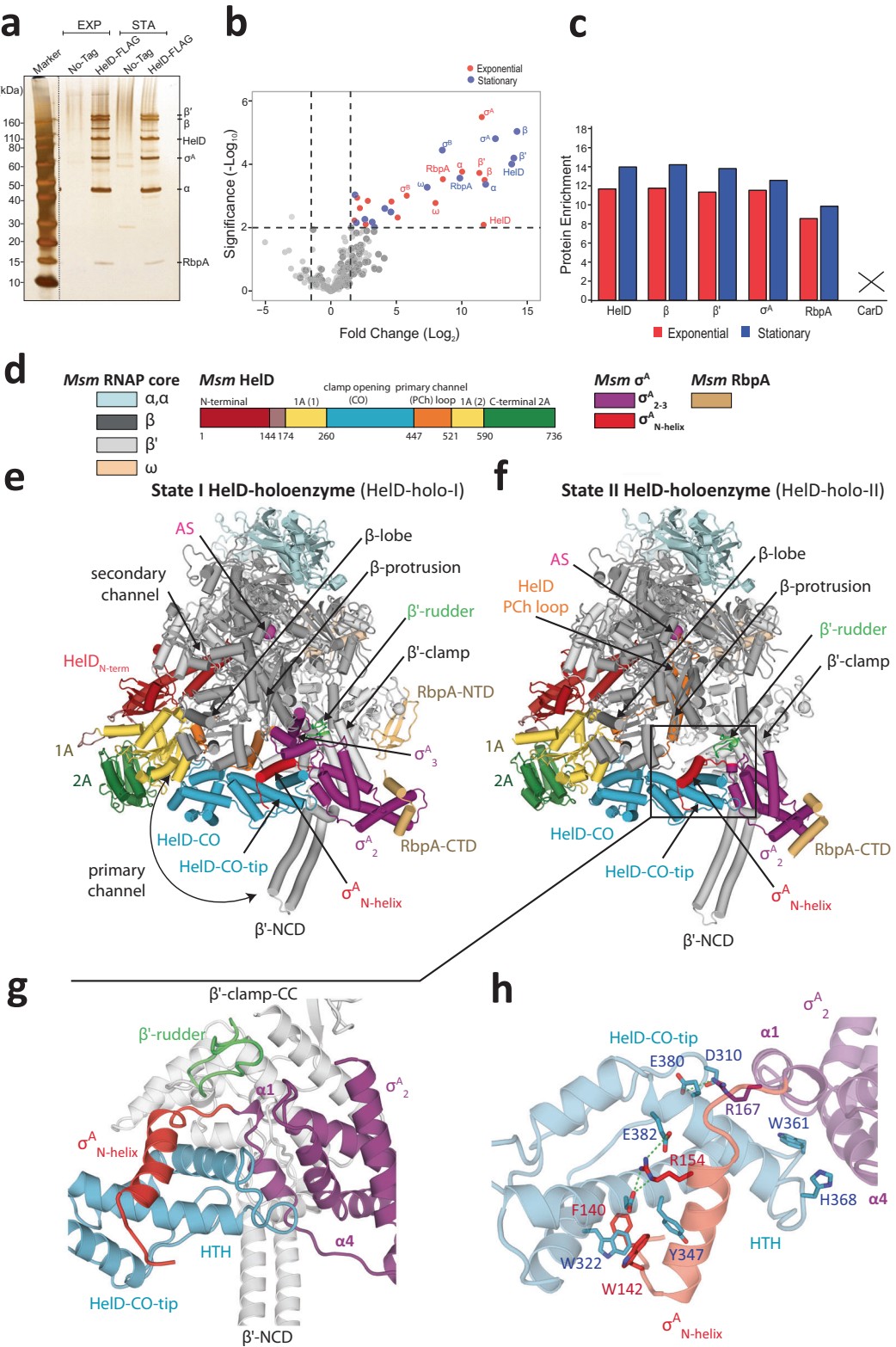

complex, the contact region of the CO-domain and β′-clamp becomes better defined, illustrating that σ$^A$ is required for the formation of a mutually stable interaction at the tip of the CO domain.

In the case of HelD–holo-I, σ$^A_3$ is in contact with the β-protrusion domain and can be visualized (Fig. 1e). However, in HelD-holo-II (Fig. 1f), the orientation of the HelD CO-domain and PCh-loop makes the RNAP clamps extremely open (Supplementary Table 3). This leads to a loss of contact between σ$^A_3$ and the β-protrusion, rendering σ$^A_3$

mobile and invisible in the structure. σ$^A_4$ is not ordered in either of the structures.

## σ$^B$ displays a less extensive interface with HelD than σ$^A$

To provide structural insight into the observed interaction of HelD and σ$^B$ in the context of RNAP (Fig. 1b), we compared *Mycobacterium tuberculosis (Mtb)* σ$^B$ (PDB 7PP4[31] [https://doi.org/10.2210/pdb7PP4/pdb]) with *Msm* σ$^A$ in the context of the HelD–holo-II complex.

**Fig. 1 | Associating partners of HelD in vivo and structure of the *Msm* RNAP core together with σ$^A$, RbpA and HelD. a** Silver-stained SDS-PAGE of HelD-FLAG pull-down from exponential (EXP) and stationary (STA) phase of growth. A No-Tag strain was used as a control. Proteins pulled down with HelD are indicated on the right-hand side. The experiment was performed four times and a representative gel is shown. The dotted line shows electronic assembly of the gel. **b** Quantitative mass spectrometry analysis of HelD-FLAG pull-down vs No-Tag strain in EXP and STA phases of growth, respectively. The analysis was done from three biological replicates. The abundance of individual proteins was compared by two-tailed student's *t*-test. The permutation-based FDR was used as an adjustment of *p*-value. The enrichment is shown with a volcano plot ($-\log_{10} p$ value > 2 on the *y*-axis, protein enrichments > 1.5 on the *x*-axis). Significantly enriched proteins are shown as red (EXP) and blue (STA) dots, respectively. The identity of the most enriched proteins is indicated. **c** Enrichment of selected proteins from (**b**) (EXP and STA) related to the transcription machinery, showing relative enrichment of the proteins in the HelD-FLAG pull-down. CarD was not present in the HelD-FLAG pull-down dataset. This is indicated with the cross. Source data are provided as a Source Data file. **d** Color-coded annotation of *Msm* RNAP core, domains of HelD, σ$^A$ and RbpA. **e, f** Two conformations of the *Msm* RNAP core complex together with σ$^A$, RbpA and HelD in state I (HelD-holo-I) and state II (HelD-holo-II), respectively. Individual domains are color-coded according to (**d**). **g** Magnified details of panel (**f**). The mutual interaction of σ$^A$ and HelD in the context of the β'-clamp. σ$^A_2$ interacts with the conserved binding site on the β'-clamp coiled-coil domain (β'-clamp CC, gray) near the β'-clamp rudder (green). The σ$^A_{N-helix}$ and adjacent regions (red) wrap with specific protein–protein interactions around the HelD–CO-tip helix-turn-helix (HTH) motif (light blue). The HelD–CO-tip is also buttressed by helices α1 and α4 of the σ$^A_2$ domain (purple). **h** Magnified details of panel (**g**). Specific residues important for the σ$^A$–HelD interaction are highlighted. σ$^A$/Phe140 (red) and its interaction with HelD defines the beginning of ordered regions of σ$^A$.

The superposition was centered at the *Msm* σ$^A_2$ domain (Supplementary Fig. 5). Helices 1 and 4 of σ$^B_2$ were compatible with the HelD CO-domain tip interaction. However, σ$^B$ lacks an equivalent of the σ$^A_{N-helix}$ to specifically wrap around the CO-domain HTH motif, resulting in the absence of this interaction interface.

## HelD is compatible with initial promoter DNA interaction

In the canonical transcription initiation pathway, σ$^A$ initially enables binding of the promoter outside the primary channel, in the so-called closed complex[32]. In order to investigate whether closed complex formation displaces HelD from HelD–holo-I and -II complexes, we reconstituted the HelD–σ$^A$–RbpA–RNAP complex with a previously used model upstream fork (us-fork) promoter DNA scaffold containing −35 and −10 elements[30] (Fig. 2a). SDS-PAGE of the SEC analysis (Supplementary Fig. 2c) revealed that binding of the us-fork DNA scaffold did not exclude HelD from the RNAP complex, indicating that it can bind RNAP simultaneously with HelD. This was also manifested by an increased UV A260/A280 ratio of SEC elution fractions containing the HelD–σ$^A$–RbpA–RNAP–DNA complex in comparison to the DNA-free sample (Supplementary Fig. 2b). The subsequent cryo-EM analysis of this complex revealed three major 3D classes (Supplementary Figs. 6 and 7).

The first class (hereafter us-fork–HelD–RPc-II), at an overall resolution 3.45 Å, visualized complete HelD bound to RNAP together with σ$^A_{N-helix}$, σ$^A_{2-4}$, both RbpA N- and C-terminal domains and the full us-fork promoter (Fig. 2b, Supplementary Movie 2). According to the HelD configuration in the primary channel, the us-fork–HelD–RPc-II resembles the HelD–holo-II complex where both the HelD CO-domain and PCh-loop occupy the primary channel.

The second class (hereafter us-fork–HelD–RPc-III), at an overall resolution 3.49 Å, visualized the HelD$_{N-term}$, the HelD PCh-loop tip and low-resolution contours of the CO-domain bound to the RNAP in a tilted orientation when compared to state I and II complexes. σ$^A_{2-4}$, the RbpA N-terminal tail, both RbpA N- and C-terminal domains, and the full us-fork promoter are also localized (Fig. 2c, Supplementary Movie 2). Density for the NTPase domain is missing in the cryo-EM reconstruction; the domain is probably mobile on the periphery of the primary and secondary channels. The CO-domain tilt results in the CO-tip losing its interaction with the σ$^A_2$ domain and σ$^A_{N-helix}$. Concomitantly, the decrease in the push against the RNAP β'-clamp allows its partial closing (compare us-fork-HelD-RPc-II and -III in Supplementary Table 3). Notably, the PCh-loop tip still occupies the active site cavity and prevents the RNAP β'-clamp from closing completely, similarly as in the previously identified HelD–RNAP state III complex[9].

The third class (hereafter us-fork–HelD$_{N-term}$–RPc-III), at an overall resolution 3.44 Å, (Fig. 2d, Supplementary Movie 2) contains HelD$_{N-term}$ in the secondary channel but lacks both the CO-domain and the PCh-loop bound in the RNAP primary channel. Release of the PCh-loop then allows further β'-clamp closure. However, the presence of HelD$_{N-term}$ in the secondary channel still restricts a complete β'-clamp closure in contrast to the previously observed us-fork promoter−σ$^A$−RNAP complex (PDB 5TW1[30] [https://doi.org/10.2210/pdb5TW1/pdb], Supplementary Fig. 8b, d). In summary, the three presented structures of HelD-containing complexes illustrate how the progressive expelling of HelD domains from the primary channel results in sequential β'-clamp closure (Fig. 2e–g: black scale bar, Supplementary Table 3).

From the point of view of σ$^A$ domains, when comparing the state II complexes with and without the us-fork promoter, the presence of the DNA enables ordering of σ$^A_{2-4}$ on the −10 and −35 elements in the same manner as observed in the us-fork promoter σ$^A$–RbpA–RNAP complex (compare Fig. 1f and Supplementary Fig. 8a, b). However, in us-fork–HelD–RPc-II, the presence of the HelD CO-domain keeps the β'-clamp swung out and the primary channel widely open. As a consequence, a large gap opens between the β-lobe/protrusion and β'-clamp and the linker between σ$^A_3$ and σ$^A_4$ becomes disordered. Concomitantly, the σ$^A_{N-helix}$, which binds along the β-lobe in the us-fork promoter σ$^A$–RNAP primary channel closed complex (Supplementary Fig. 8d), is wrapped around the HelD CO-domain in us-fork–HelD–RPc-II (Supplementary Fig. 8c). Interestingly, the σ$^A_{N-helix}$ has been previously identified also across the primary channel in RNAP–σ$^A$ holoenzyme[33]. Comparison of these structures illustrates how this mobile element of σ$^A$ modulates its location in response to the conformational status of the whole enzyme.

## HelD N-terminal domain is expelled from the secondary channel on the way to a transcription initiation intermediate

To form the so-called open complex where the transcription bubble is established, RNAP follows a multistep process[34,35] during which it loads the DNA promoter into the primary channel, opens the transcription bubble and completely closes the polymerase clamp around it. Indeed, HelD must be released from the primary channel to allow open complex formation. To visualize the process of HelD release, we reconstituted the HelD–σ$^A$–RbpA–RNAP complex with full DNA promoter sequence with an artificially opened transcription bubble[30] (Fig. 3a). Such a reconstituted complex was then immediately frozen on cryo-EM grids and analyzed (Supplementary Figs. 9 and 10). The major 3D class, at an overall resolution 3.09 Å, was a canonical open complex (RPo) without HelD. The structure is very similar to the *Msm* transcription initiation complex with a full transcription bubble (PDB 5VI5[30] [https://doi.org/10.2210/pdb5VI5/pdb]) except for the absence of an RNA product in the active site cavity. Additionally, two minor 3D classes were also revealed by the cryo-EM analysis.

The first minor class (hereafter HelD$_{N-term}$–RP2 Fig. 3b, Supplementary Movie 3), at an overall resolution 3.16 Å, captured a transcription initiation intermediate where HelD$_{N-term}$ is still loosely bound to the secondary channel, represented by a blurred density (Supplementary Figs. 9 and 10). The complex resembles the CarD-stabilized

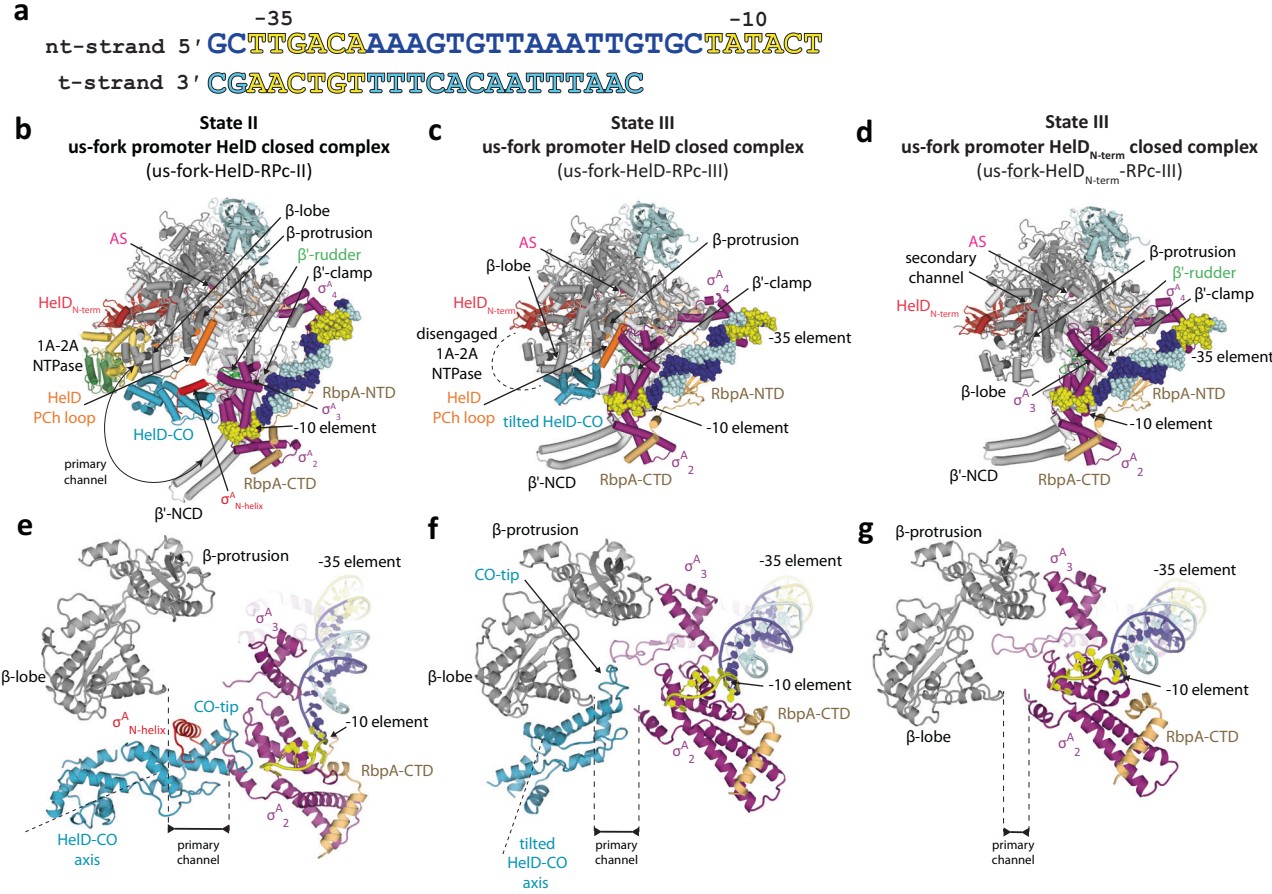

**Fig. 2 | DNA upstream fork promoter binds to *Msm* HelD–σ$^A$–RbpA–RNAP complex. a** Sequence of the us-fork promoter DNA fragment. The numbers above denote the DNA position with respect to the transcription start site (+1). The −35 and −10 elements are colored yellow, nt/t denotes non-template/template strand, respectively. **b–d** Three conformations of the *Msm* RNAP core complex together with us-fork promoter DNA fragment, σ$^A$, RbpA, and HelD. One conformation is in state II (us-fork–HelD–RPc-II) and two conformations are in state III (us-fork–HelD–RPc-III, us-fork–HelD$_{N-term}$–RPc-III), respectively. In us-fork–HelD–RPc-II, the whole HelD protein is ordered on RNAP, in us-fork–HelD–RPc-III the 1A–2A NTPase is disengaged and thus HelD–CO tilts relative to the primary channel. In us-fork-HelD$_{N-term}$-RPc-III, only the HelD$_{N-term}$ domain is bound in the secondary channel and the rest of the HelD protein is not ordered.

Individual domains are color-coded as defined in Fig. 1d. **e–g** Close-up views of the RNAP primary channel, corresponding to panels (**b–d**), respectively. The black scale bar illustrates the distance between the β-lobe and the N-terminus of the σ$^A_2$ domain, which directly correlates with the primary channel closure according to Supplementary Table 3. **e** Presence of HelD–CO (light blue) keeps the RNAP primary channel wide open. The σ$^A_{N-helix}$ wraps specifically around the HelD–CO-tip. **f** Tilting (compare HelD–CO axes) of HelD–CO disfavors CO-tip interaction with the σ$^A_2$ domain and prevents CO-tip interaction with the σ$^A_{N-helix}$. **g** Displacement of all HelD domains, except for HelD$_{N-term}$(as depicted in **d**), allows a partial closure of the RNAP primary channel but not to the extent that would allow the σ$^A_{N-helix}$ interaction with the β-lobe as seen in the σ$^A$–RbpA–RNAP complex (Supplementary Fig. 8d).

*Mtb* RP2 transcription initiation intermediate (Fig. 3d, PDB 6EE8[34] [https://doi.org/10.2210/pdb6EE8/pdb]) where dwDNA is partially loaded into the primary channel and clamped in between the β-lobe and β′-clamp. In RP2, the dwDNA interaction with fork-loop 2 (FL2) and switch 2 (Sw2) blocks dwDNA to move fully into the primary channel (Supplementary Fig. 11c), which has been previously identified as a universal regulatory step in transcription initiation[34,36]. There is also only a narrow gap in between FL2 and the Sw2 that prevents the single-strand template DNA from reaching the AS, and thus the transcription bubble is not yet fully formed. In HelD$_{N-term}$–RP2, the HelD$_{N-term}$ presence partially prevents the β′-clamp closure, resulting in a slightly more open conformation than in the CarD RP2 complex (Supplementary Table 3). This causes that the dwDNA is neither engaged with the β-lobe nor with FL2 and Sw2 (Supplementary Fig. 11a), and only loosely buttressed by the β′-clamp, resulting in a blurred density for the dwDNA duplex partially loaded into the primary channel. The melted template strand (t-strand) and a part of the t-strand −10 discriminator (the region between −10 and +1, the transcription start site [TSS]) are completely disordered. On the other hand, the melted non-

template (nt) strand is securely wrapped around σ$^A_{1.2}$ and σ$^A_2$ domains, maintaining canonical interactions. Overall, the HelD$_{N-term}$-RP2 structure represents a transcription initiation intermediate where HelD$_{N-term}$ binding in the secondary channel is compatible with partial promoter melting and partial loading of the dwDNA into the primary channel. As a result, the concomitant closure of the RNAP clamp causes a movement of the β′-jaw constituent and of the trigger loop-bearing region of the secondary channel towards the RNAP AS (Supplementary Fig. 12). HelD$_{N-term}$ maintains its interactions with the opposite wall of the secondary channel (the β′-funnel helices) and impinges on the bridge helix. This prevents HelD$_{N-term}$ movement with the closing of the primary channel and so HelD$_{N-term}$ gradually loses contacts with the β′-jaw and trigger loop side of the channel. In this way, the progressive closure of RNAP leads to deterioration of the binding site for HelD$_{N-term}$.

In the second minor 3D class (hereafter σ$^A_{N-helix}$–RP2, Fig. 3c, Supplementary Movie 3), at an overall resolution 3.89 Å, HelD is no longer present. Nevertheless, there is an interpretable density of the σ$^A_{N-helix}$ situated in between the β-lobe and β′-clamp domains.

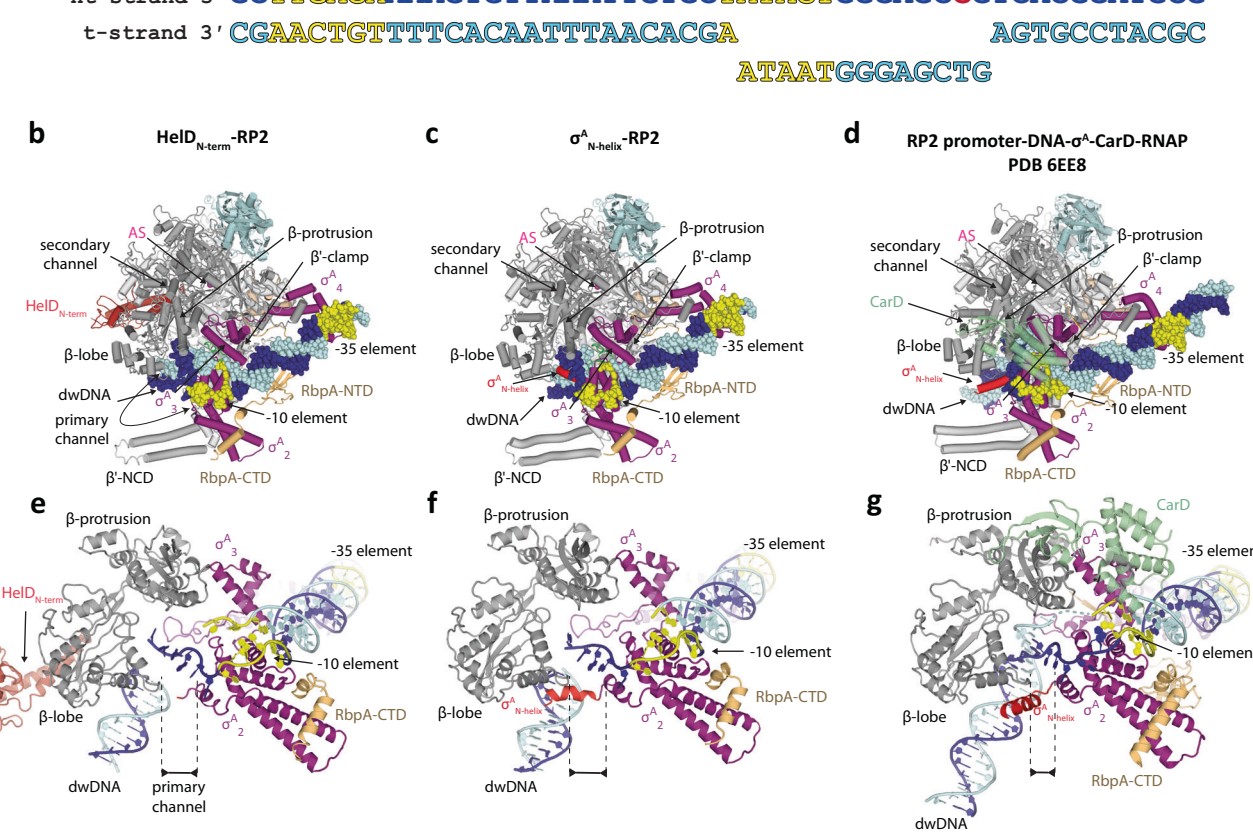

**Fig. 3 | HelD release on the pathway towards RPo complex formation.**
**a** Sequence of the promoter transcription bubble DNA fragment. The numbers above denote the DNA position with respect to the TSS (+1). The −35 and −10 elements are colored yellow; nt/t denotes non-template/template strand, respectively. **b** *Msm* RNAP core complex together with the promoter transcription bubble DNA fragment, σ$^A$, RbpA and HelD in the RP2 like-state (HelD$_{N-term}$-RP2). Only the HelD$_{N-term}$ domain is present in the secondary channel, the rest of the HelD protein is not ordered. The dwDNA is partially loaded into the primary channel. Individual domains are color-coded as in Fig. 1d. **c**, **d** *Msm* RNAP core complex together with the promoter transcription bubble DNA fragment, σ$^A$, RbpA, (no HelD), in the RP2-like state (σ$^A_{N-helix}$−RP2) and *Mtb* RP2 RNAP complex (RP2 promoter−DNA−σ$^A$−RNAP) PDB 6EE8, respectively. Individual domains are color-coded as in Fig. 1d, CarD in panel **d** is

transparent green. **e–g** Close-up views of the RNAP primary channel from panels (**b**–**d**), respectively. The black scale bar illustrates the distance between the β-lobe and the N-terminus of the σ$^A_2$ domain, which directly correlates with the primary channel closure according to Supplementary Table 3. **e** The presence of HelD$_{N-term}$ (firebrick) in the secondary channel prevents the RNAP primary channel from closing completely. Concomitantly, dwDNA is only partially loaded into the primary channel. **f** Displacement of the HelD$_{N-term}$ domain is followed by a slight adjustment of the RNAP primary channel and interaction of σ$^A_{N-helix}$ with the dwDNA. **g** In the RP2 complex (PDB 6EE8), the RNAP primary channel closes around dwDNA so that the σ$^A_{N-helix}$ directly interacts with the β-lobe domain. CarD interacts with the −10 element and stabilizes the transcription bubble.

However, the low local resolution precludes identification of any specific interaction among the proteins. The σ$^A_{N-helix}$ is positioned in a manner that it provides support to the dwDNA duplex being loaded into the primary channel. Indeed, there are three positively charged residues of the σ$^A_{N-helix}$ which might interact with the DNA phosphate backbone. When comparing the σ$^A_{N-helix}$-RP2 and HelD$_{N-term}$-RP2 structures, the dwDNA duplex itself and the rest of the DNA promoter are in a very similar configuration (Supplementary Fig. 11b). However, there is a slight twist[37] in the β′-clamp.

Taken together, the HelD$_{N-term}$−RP2 and σ$^A_{N-helix}$−RP2 structures enabled us to visualize the sequential process of HelD$_{N-term}$ release from the secondary channel on the pathway to a fully open promoter complex. We note that HelD$_{N-term}$−RP2 and σ$^A_{N-helix}$−RP2 are not equivalents of the CarD RP2 intermediate (Fig. 3e–g) with respect to RNAP β′-clamp opening and promoter loading configuration.

### Effect of HelD on CarD binding to RNAP
Our pull-down experiments suggested that CarD is not present on RNAP together with HelD. CarD consists of two domains[23]. The

N-terminal domain (termed RID) interacts with the RNAP β-lobe[38]. The C-terminal domain interacts with the −10 element of the promoter DNA at the us-fork of the transcription bubble[39]. We compared our cryo-EM data of us-fork−HelD−RPc-II and -III with PDB 4XLS[39] [https://doi.org/10.2210/pdb4XLS/pdb] (us-fork−CarD−σ) that shows binding of CarD to the RNAP−σ$^A$ holoenzyme in the presence of DNA. In the two HelD complexes in comparison with us-fork−CarD−σ, the results revealed a large relocation of the promoter −10 element together with σ$_2$ by ~39 Å or 5.4 Å away from the β-lobe, respectively (Supplementary Fig. 13), depriving CarD of a crucial binding interface. This likely weakens CarD binding, explaining the lack of simultaneous presence of CarD and HelD on RNAP (Fig. 1c, Supplementary Tables 1 and 2).

### Effect of NTP binding/hydrolysis on HelD release from RNAP
Our structural experiments presented in this study demonstrated that class II *Msm* HelD can be released in the absence of ATP, as a result of conformational changes induced by σ$^A$ and DNA interactions with RNAP taking place during transcription initiation. Nevertheless, previous results indicated that the release of class I *Bacillus subtilis* HelD

from RNAP might be stimulated by a non-hydrolyzable ATP analog[18]. However, our attempts to visualize ATP or non-hydrolyzable ATP analog bound to *Msm* HelD in complex with RNAP failed. Indeed, the binding site of the NTP base in HelD−RNAP structures inferred from comparison with a UvrD complex[40], was occluded by HelD/Tyr589−Arg590 of the NTPase active site. Likewise, the NTP-binding pocket in the presented structures was not in an NTP binding-compatible conformation.

To start characterizing the role of ATP/GTP in HelD release from RNAP, we first measured the ATPase/GTPase activities of both free HelD and HelD in complex with the RNAP holoenzyme. Interestingly, the ATPase activity measured for the HelD−core and HelD−holoenzyme complexes was ~2.5-fold higher than the activity of free HelD (Fig. 4a) while almost no difference was observed for GTPase activity (Fig. 4b). This suggests that the HelD ATP hydrolysis but not GTP hydrolysis is stimulated in the context of the RNAP holoenzyme.

To further clarify the effect of ATP and also explore the potential effect of other NTPs for release of class II HelD from RNAP, we assembled the *Msm* RNAP−σ^A−RbpA−HelD complex where RNAP was attached via its His-tag to cobalt-based magnetic beads (Fig. 4c). We then incubated the complex either with ATP or GTP or CTP or in the buffer only. HelD released to the supernatant was then visualized on SDS-PAGE and quantified (Fig. 4d, Supplementary Fig. 14). The results showed some HelD release even in the absence of NTPs. The release was then markedly stimulated by ATP, less by GTP, and CTP had no stimulatory effect. The stimulation by ATP was not concentration-dependent and remained almost unchanged between 1 and 8 mM (Supplementary Fig. 15). Due to the most prominent effect of ATP on HelD release at 1 mM concentration, we used it in subsequent experiments.

We then wished to test the effect of ATP binding and/or hydrolysis on HelD release. We created two mutant variants of HelD, HelD^A-HYDRO, and HelD^A-BIND (Supplementary Fig. 16). In HelD^A-HYDRO, specific amino acid residues were mutated to abolish ATP hydrolysis but not binding. In HelD^A-BIND, ATP binding (and thus also hydrolysis) was abolished. Supplementary Fig. 17 shows that both mutants were defective in ATP/GTP hydrolysis. Binding of wild type (WT) and mutated HelD variants to RNAP was about the same in our experimental setup (Supplementary Fig. 18; no statistically significant difference). We subsequently evaluated release of WT−HelD and the two mutant versions from RNAP by ATP, N-ATP (non-hydrolyzable analog), and ATPγS (analog with decreased hydrolysis potential). Figure 4e shows that WT−HelD was about equally efficiently dissociated from RNAP with ATP and ATPγS. Dissociation by N-ATP was reduced compared to ATP but only by about 50%, suggesting that binding of ATP itself contributes to HelD release. HelD^A-HYDRO showed reduced release by all three compounds. The reduction was most prominent with N-ATP, perhaps reflecting potentially compromised binding of this ATP analog to the mutant form of HelD. Finally, HelD^A-BIND was not released from RNAP by any of the compounds.

We concluded that ATP, and to a lesser degree also GTP, stimulated HelD release from RNAP. Both ATP binding and hydrolysis contributed to the release.

### Effect of σ^A and/or RbpA on HelD release from RNAP
Next, we evaluated the effect of σ^A and RbpA on HelD release from RNAP. We used the same experimental setup as in the previous experiments. We assembled complexes of RNAP with HelD alone or in combination with σ^A and/or RbpA. We verified that equal amounts of HelD were bound to RNAP (Supplementary Fig. 18). HelD release was then induced with ATP. Figure 4f shows that HelD release was not affected by these factors or their combination. To corroborate this finding, we created a variant of HelD where the HelD−σ^A interface was disturbed (HelD^σA-INT, Supplementary Fig. 16). Consistently, the release of this variant from RNAP was comparable to that of WT−HelD (Fig. 4e).

### Effect of DNA on HelD release from RNAP
Subsequently, using the RNAP−σ^A−RbpA−HelD complex, we tested how two forms of DNA, one mimicking the closed complex (CC), the other mimicking the open complex (OC), affect HelD release from RNAP. Figure 4g shows that CC DNA on its own had no discernible effect, consistent with the structural data. Addition of ATP and CC DNA, however, appeared to have a moderately more pronounced effect than ATP alone. OC DNA then stimulated the release even in the absence of ATP, again correlating with our structural analysis. Finally, the combination of OC DNA and ATP was more efficient than either of these two components alone, and also more than a sum of the two effects, suggesting their synergy.

### Effect of HelD on transcription
Finally, we characterized the effect of HelD on transcription in a defined in vitro system. We performed multiple round transcriptions in the absence or with increasing amounts of HelD from the P*rrnAPCL1* ribosomal RNA (rRNA) promoter[41] with RNAP complexed with σ^A (Supplementary Fig. 19a) and increasing levels of RbpA (Supplementary Fig. 19b). The results were consistent with the known stimulatory effect of RbpA and revealed that increasing amounts of HelD decreased the overall yield of transcription, correlating with its ability to sequester RNAP.

Next, as HelD also binds into the secondary channel of RNAP we asked whether HelD affects the affinity of RNAP for iNTP, which reflects the stability of the RPo. Promoters that form relatively unstable RPos, such as rRNA promoters, can be regulated by changes in the concentration of iNTP[42–44] and this regulation is potentiated by factors that bind into the secondary channel of RNAP. Mechanistically, binding of these factors to the secondary channel coincides with RPo formation and influences the RPo stability on those regulated promoters. This, in turn, is translated into sensitivity to concentration of iNTP−less stable RPos require relatively higher concentrations of iNTP for maximal transcription than more stable RPos[45]. Examples of such factors are the *E. coli* proteins DksA[46] and TraR[37]. To test this hypothesis, we performed transcription in vitro with the P*rrnAPCL1* rRNA promoter as a function of increasing concentration of iNTP. Supplementary Fig. 20 shows that regardless of the presence/absence of HelD, the concentration of iNTP required for half-maximal transcription remained relatively unchanged, suggesting that HelD does not affect the affinity of RNAP for iNTP in a manner similar to that of, e.g., DksA. This is consistent with the structural data observation where HelD binding to the secondary channel does not coincide with RPo formation.

Finally, the presence of HelD on RNAP during the first steps of transcription initiation suggested that this presence may be beneficial with respect to rifampicin resistance. Protective effects of HelD had been shown for *M. abscessus* and *S. venezuelae* RNAPs[13,14], and we wished to ascertain whether *Msm* HelD possessed the same property. Multiple round transcriptions in the presence/absence of HelD and/or increasing levels of rifampicin demonstrated that transcription was less inhibited by the antibiotic in the presence of HelD (Fig. 5a). Figure 5b then shows a close-up view of the rifampicin binding pocket in the presence of HelD during transcription initiation, revealing how the presence of HelD distorts the binding site of the antibiotic.

## Discussion
In this study, we describe the intricate interplay between *Msm* HelD and RNAP during the transcription cycle, focusing in detail on its functioning during transcription initiation. HelD associates with RNAP, either with the core enzyme as shown in previous studies where HelD was demonstrated to dissociate stalled EC complexes[9], or with the RNAPσ^A or σ^B holoenzyme and RbpA where it is involved in transcription initiation (Fig. 6). After the EC disassembly, HelD can either be released from RNAP, which is promoted by the action of ATP or GTP, or stay on it. After σ^A and RbpA bind to RNAP−HelD, a quaternary

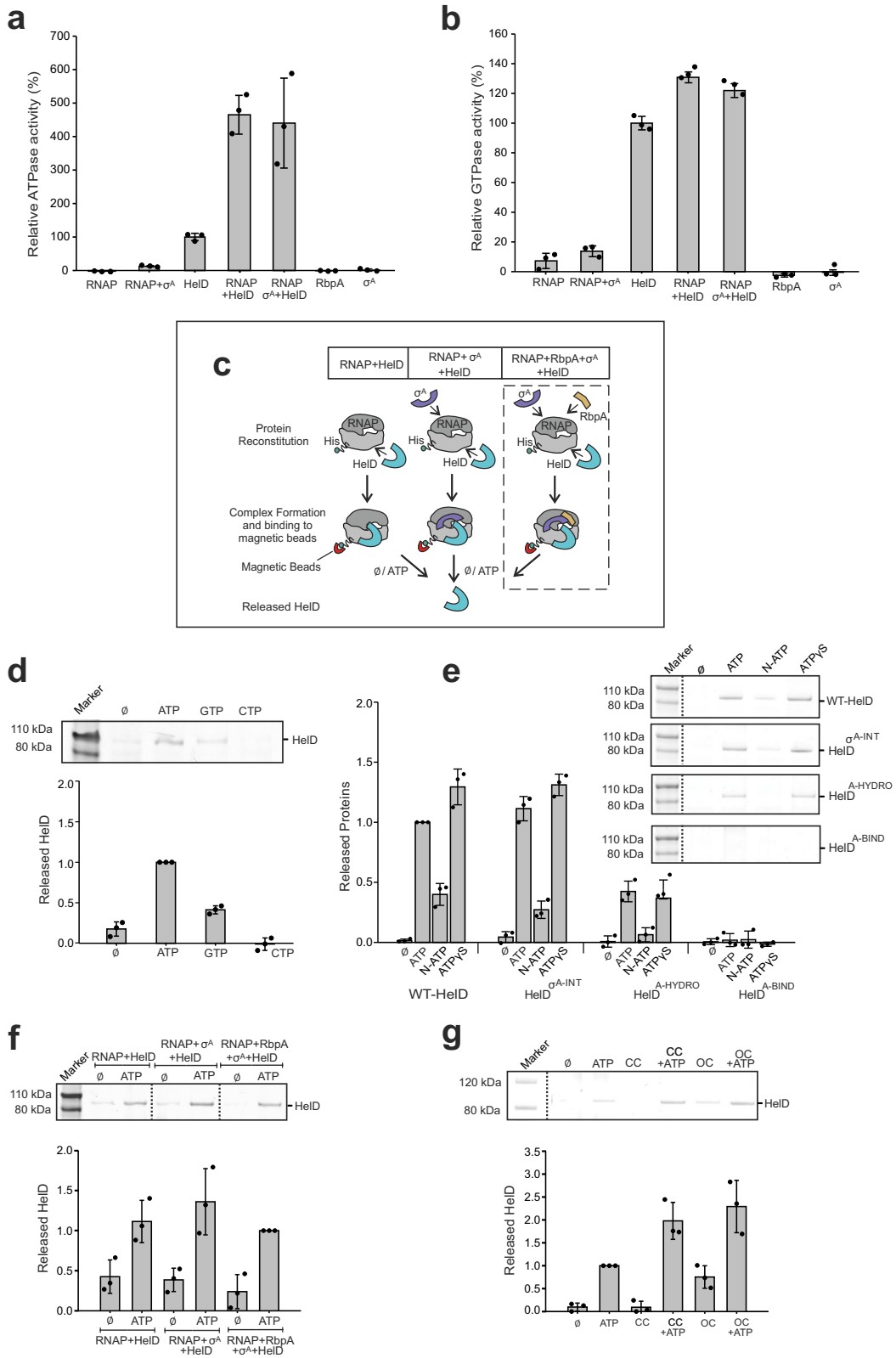

complex is formed. The *Msm* HelD–σ<sup>A</sup>–RbpA–RNAP complex (Fig. 6a) is then competent for the initial interaction with the promoter DNA positioned outside the primary channel in a closed complex-like fashion (Fig. 6b). During subsequent isomerization towards RPo formation (Fig. 6c–g), HelD is gradually released in defined steps. This process can be stimulated by ATP/GTP binding/hydrolysis. Taken together, the association of HelD with RNAP can last from the disassembly of the stalled EC until the start of the next round of

transcription, linking the two processes, and, additionally, playing a role in protecting RNAP against rifampicin (Fig. 6b, c). However, for transcription to start, HelD must fully dissociate.

## Dissociation of HelD from RNAP
To dissociate, HelD must exit both the RNAP primary and secondary channels (Supplementary Movie 4) to allow for complete DNA loading and full transcription bubble formation within RPo (Fig. 6g), and to

**Fig. 4 | NTPase activities of HelD and release of HelD from RNAP.**
**a**, **b** Comparison of NTPase activities of free HelD and its complexes with RNAP. ATP/GTP hydrolyzing activity of free HelD was set as 100%. ATP hydrolysis (**a**) is stimulated upon complex formation, whereas GTP hydrolysis (**b**) remains almost unchanged. Control measurements for individual complex components are shown. The bars show averages from three biological replicates, the error bars are ±SD, the dots represent individual experiments (also in panels **d**–**g**). **c** A scheme depicting the HelD release assay: His–RNAP was reconstituted into three different complexes, each containing combinations of HelD (cayn), σ$^A$ (purple) and RbpA (yellow). The RNAP complexes were then allowed to bind to magnetic beads. The amount of HelD released, with or without addition of other factors (in panels **d**–**g**) was determined by Coomassie blue-stained SDS-PAGE gels and densitometry. **d** Effect of 1 mM ATP, GTP, or CTP on HelD release. In panels **d**–**g**, representative primary data are shown above the graph. Zero (∅) shows HelD release without the addition

of other factors. For this and experiments (**e**, **g**), the His–RNAP complex containing HelD, σ$^A$, and RbpA was used (depicted within the dashed box in **c**). The amount of HelD released from RNAP–σ$^A$–RbpA–HelD by the addition of ATP was set as 1 (also in other panels). A second primary data example is shown in Supplementary Fig. 14 together with a calibration curve used as quantification control. **e**, Effect of ATP analogs on HelD release. RNAP complexes were reconstituted as described in panel **c** with four HelD variants: WT-HelD (wild type), HelD$^{σA-INT}$, HelD$^{A-HYDRO}$, and HelD$^{A-BIND}$ (for definition of the mutants see Supplementary Fig. 16). Subsequently, 1 mM each of ATP, N-ATP, or ATPγS was added to the preformed RNAP complex attached to the magnetic beads and release of HelD from the complex was observed. **f** Release of HelD from the three types of complexes (**c**) induced with 1 mM ATP. **g** Effect of two forms of DNA and/or ATP on HelD release. CC, closed complex us-fork promoter DNA. OC open complex DNA with artificially opened transcription bubble. Source data are provided as a Source Data file.

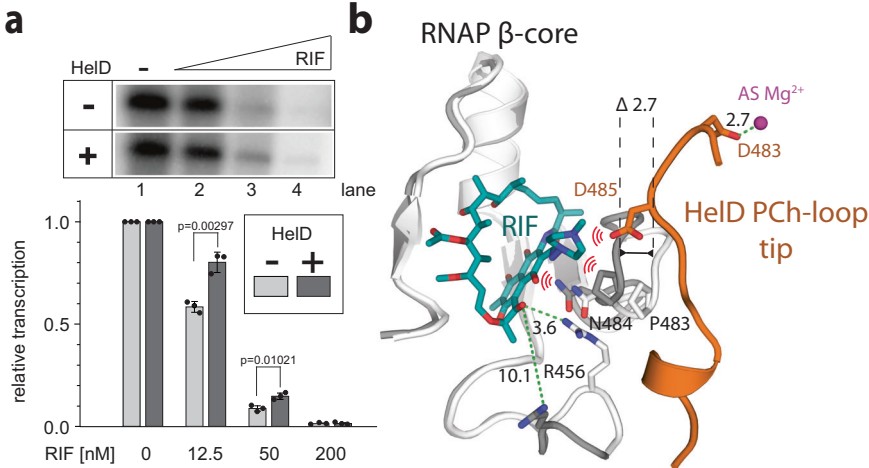

**Fig. 5 | Protective effect of HelD against rifampicin. a** Multiple round transcriptions from the *Msm* rRNA P*rrnAPCL1* promoter were performed in the absence or presence of HelD with increasing amounts of rifampicin (RIF). The 1:1 RNAP:HelD ratio was used in protein reconstitution. Transcription at zero RIF was set as 1 for both ±HelD to facilitate visualization of the changes. The relative transcription in the absence of RIF and the presence of HelD compared to the absence of HelD was 72.4 % (lane 1). The bars show averages of three independent experiments, the dots are individual experimental data, the error bars show ±SD. *p* Values were calculated using a two-tailed, unpaired *t*-test. Source data are provided as the Source Data file.

**b** HelD primary channel (PCh) loop binding causes conformational changes in the RNAP rifampicin binding site. *Mtb* RNAP–rifampicin complex (RIF in teal, RNAP β-core in light gray, PDB 5UHC) is superposed with the *Msm* us-fork–HelD–RPc-II complex (dark gray, active site Mg$^{2+}$ in pink). Binding of the *Msm* HelD PCh loop (orange) deforms the RIF binding pocket: β-core/P483–N484 loop is pushed towards RIF by 2.7 Å, D485 of the HelD PCh loop itself sterically clashes with RIF, and β-core/R456, which usually coordinates RIF, is moved away. Possible atomic clashes between RIF and its deformed binding site and HelD are hinted with red 'wave' symbols, distances (green dashed lines) are in Å.

allow the constituents of the AS to catalyze the first nucleotidyl transfer reaction. In the case of *Msm* HelD, particularly the CO domain and the PCh-loop, which directly interacts with the AS Mg$^{2+}$ ion, and HelD$_{N-term}$, which restricts the trigger loop folding to the catalytically permissive conformation, must vacate RNAP. Our structural analyses illustrate that in the first phase of HelD displacement, the CO domain and the associated 1A–2A NTPase domain are expelled from the primary channel while the PCh-loop and HelD$_{N-term}$ still remain in the primary and secondary channels, respectively. The HelD$_{N-term}$–RP2 structure then represents an intermediate in the HelD release process where partial melting and loading of the dwDNA expels the PCh-loop while HelD$_{N-term}$ remains in the secondary channel (Fig. 6e). During the HelD release process, the RNAP β'-clamp progressively closes, propagating conformational changes that ultimately reach the secondary channel, gradually disfavoring HelD$_{N-term}$ binding. The trigger loop, folded in a catalytically non-permissive conformation in State II and State II-like complexes, becomes disordered upon PCh-loop leaving the primary channel. This results in HelD$_{N-term}$ losing interactions with this part of the trigger loop. As a consequence, HelD$_{N-term}$ leaves the secondary channel, as seen in the σ$^A_{N-helix}$–RP2 structure (Supplementary Fig. 12).

In the initial *Msm* HelD–σ$^A$–RbpA–RNAP complex, σ$^A_2$ and σ$^A_{N-helix}$ extensively interact with the tip of the HelD CO domain. We tested the importance of this interface during release of HelD from RNAP and did not detect any significant differences between WT–HelD and the HelD–σ$^A$ interface mutant HelD (HelD$^{σA-INT}$); both dissociated about equally. This could be due to the extensive interface between HelD and RNAP playing the major role, suggesting that the σ$^A$–HelD interaction does not play a significant part in the HelD release process.

**Open complex formation**
In the σ$^A_{N-helix}$–RP2 structure, the further dwDNA loading into the primary channel is stabilized by its interaction with the σ$^A_{N-helix}$ (Fig. 6f). Nevertheless, the dwDNA still needs to be inserted more deeply into the primary channel[36], the transcription bubble has to propagate further to fully separate individual DNA strands, and the template strand must pass the narrow gap between FL2 and Sw2 to enter the AS cavity. Similarly, as in CarD RP2, FL2 and Sw2 in σ$^A_{N-helix}$–RP2 are probably too close to each other to allow the template strand passage. Therefore, RNAP must become temporarily open to allow the template strand transition[34].

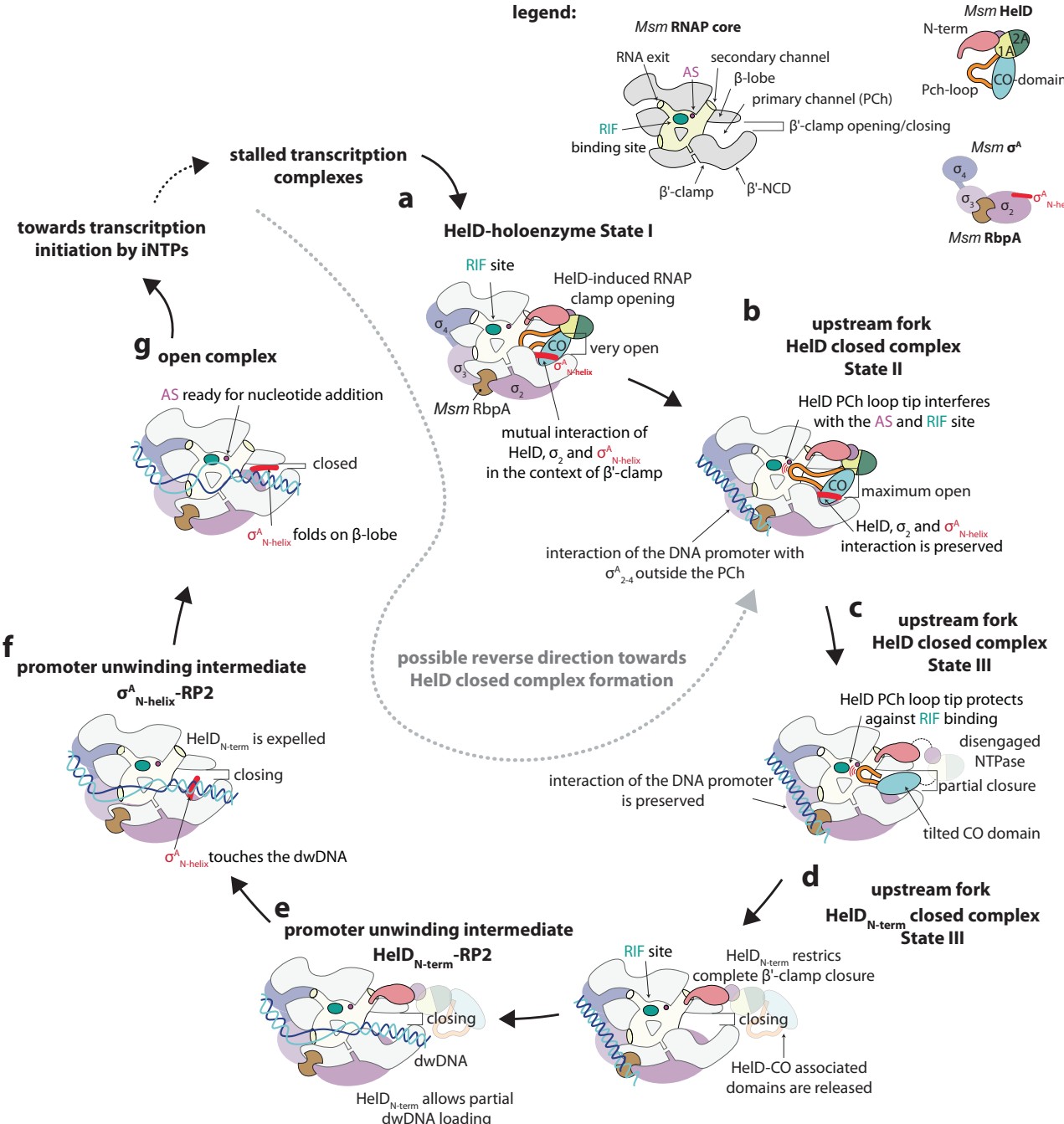

**Fig. 6 | HelD participates during transcription initiation.** The circular arrangement model of HelD participation in transcription initiation is only the best approximation of the event succession displayed in panels **a–g**: **a** HelD binds to the RNAP core together with σ^A and RbpA. Individual domains are color-coded according to the legend above. The HelD CO-domain interacts with σ^A_2 and σ^A_N-helix in the context of β′-clamp. In State I, the presence of full-length HelD in the primary channel (PCh) makes the β′-clamp wide open. However, it does not interfere with the AS and the RIF binding pocket. **b** In State II, the RNAP β′-clamp is maximally open and the HelD PCh-loop reaches the RNAP AS cavity and interferes with the RIF binding pocket. Concomitantly, in State II, the HelD–σ^A–RbpA–RNAP complex is able to recognize and bind DNA promoter outside the primary channel. **c** Disengagement of the HelD NTPase domain loosens the grip of the CO domain on the β′-clamp, the consequent narrowing of the primary channel tilts the CO domain.

Still, the PCh-loop tip is folded into to AS cavity and interferes with the RIF binding pocket. **d** HelD clearance from the primary channel allows for β′-clamp closing, however HelD_N-term presence in the secondary channel restricts the full closure. **e** Partial loading of dwDNA is compatible with HelD binding to the secondary channel, however, further interaction of dwDNA within the RNAP primary channel (**f**) triggers a conformational change of the secondary channel which disfavors and expels HelD_N-term. σ^A_N-helix helps to accommodate dwDNA towards the RP2-like intermediate. **g** In order to establish the RNAP open complex (competent for the first cycle of nucleotide addition), the DNA promoter still needs to be further accommodated into the RNAP AS cavity. At that moment, RNAP clamps around the dwDNA and σ^A_N-helix locks the clamp by interaction with the β-lobe. From this state the complex can proceed towards transcription initiation or, when stalled by RIF, reverse towards HelD–RPc (gray dashed line arrow).

In $\sigma^A_{N\text{-helix}}$–RP2, the $\sigma^A_{N\text{-helix}}$ directs the rest of the $\sigma^A$ N-terminal domain (mobile, not captured in the structure) towards the gap at the vestibule of the primary channel between the β-lobe and the coiled-coil motif of the β′-clamp non-conserved domain (β′-NCD), likely sealing it. This position of the $\sigma^A$ N-terminal domain then would favor a direct contact with dwDNA (Figs. 3f and 6f). This contrasts with CarD RP2 and RPo, where $\sigma^A_{N\text{-helix}}$ folds along the β-lobe and points the rest of the $\sigma^A$ N-terminus outside the primary channel[30] (Fig. 3g). In us-fork–HelD–RPc-II, $\sigma^A_{N\text{-helix}}$, wrapped around the HelD–CO domain (Fig. 2e), would also direct the $\sigma^A$ N-terminal domain outside the primary channel, which is occupied by the HelD–CO domain itself. However, the $\sigma^A$ N-terminal domain would be close to dwDNA where promoter nucleation occurs, and also close to β′-NCD. It is unclear, whether and how exactly $\sigma^A_{N\text{-helix}}$ and the rest of the adjacent $\sigma^A$ N-terminal domain facilitate the dwDNA loading. Nevertheless, the $\sigma^A$ N-terminal domain has the suitable spatial position to do so in the us-fork–HelD–RPc-II and $\sigma^A_{N\text{-helix}}$–RP2 complexes. Compared to the *E. coli* $\sigma^{70}$ transcription system[35], it seems the *Msm* $\sigma^A$ N-terminal domain indeed never resides in the primary channel itself[30].

We note that the HelD$_{N\text{-term}}$–RP2 and $\sigma^A_{N\text{-helix}}$–RP2 are reminiscent of the CarD RP2 intermediate during transcription initiation when HelD is present instead of CarD[34]. It is also similar to the *E. coli* TraR-assisted transcription initiation[35], where TraR binds to the secondary channel and contributes to stabilization of transcription initiation intermediates. By analogy, the HelD$_{N\text{-term}}$–RP2 and $\sigma^A_{N\text{-helix}}$–RP2 intermediate resemble the TraR pre-open complex (T-preRPo) where dwDNA is partially loaded and the $\sigma^{70}_{1.1}$ domain is just ejected from the primary channel. However, in contrast to HelD, TraR remains bound also to the fully established RPo complex and leaves only before the first nucleotidyl transfer reaction. Consistently, unlike TraR[37,47], HelD does not appear to alter the RPo stability as assayed by the requirement of RNAP for the concentration of iNTP for half-maximal transcription (Supplementary Fig. 20).

We also note that binding of HelD to RNAP seems to exclude the presence of CarD, due to abolishing its binding niche. When HelD becomes dissociated, however, CarD might assume its place and participate in the final stages of transcription initiation.

Taken together, the visualized complexes illustrate HelD-assisted transcription initiation. This classifies HelD among protein factors which are involved in the bacterial transcription initiation pathway, such as DksA[48], TraR[35], or SutA[49] although HelD appears to function differently.

## ATPase role in the HelD cycle

Our structural and biochemical experiments showed that HelD dissociation from RNAP can occur in the absence of NTPs. However, the biochemical experiments also revealed that the HelD ATPase activity is enhanced upon RNAP binding. As for release of HelD from RNAP, ATP binding alone stimulates this process and it is even further stimulated by ATP hydrolysis. Supplementary Figure 15 also suggests that HelD is perhaps not regulated by intracellular ATP concentration, as 1 mM ATP releases HelD from RNAP almost as efficiently as 8 mM ATP. The reported ATP concentration in exponential phase in *Msm* is then 4 mM[50]. Furthermore, the release of HelD is stimulated not only when ATP is added but it is even further stimulated when the RNAP complex interacts with promoter DNA, especially with the open complex. Hence, not all HelD is released by ATP alone, some is still retained by a fraction of RNAP molecules. This suggests that at least in some cases HelD may remain bound to RNAP from the moment of helping release stalled RNAPs from DNA to the moment of participating in the next round of transcription initiation.

All the structural states of *Msm* HelD complexes observed so far fall into two categories with respect to the conformational status of the NTPase active site. In State I-like structures, the overall NTPase organization compares well with the previously described NTPase

complexes with substrate analogs or reaction products in a 'closed' conformation[40]. The State I-like state would be capable of NTP binding and hydrolysis upon change of the HelD/Tyr589–Arg590 loop and a slight readjustment of the side chains of the HelD active site. However, access to the active site of HelD is barred by the NG-linker of HelD[9]. In State II-like structures, there is free access to the active site of HelD as the NG-linker region is not localized in the structure, but the catalytic HelD/Glu529 is retracted by 2.3 Å from the optimal position for the hydrolysis reaction. In State III-type of structures, the NTPase domain is not defined and the status of the active site of HelD is unknown.

The transition between State I-like and II-like is accompanied by changes in the extension of the 1A domain linked to the PCh-loop. We hypothesize that a structural change upon NTP binding and perhaps hydrolysis might help pull the PCh-loop from the primary channel and enhance the HelD release process from RNAP. However, the exact chain of events in this process is still elusive.

## HelD-assisted transcription initiation

What is the advantage of the intimate association of class II HelD with RNAP during transcription initiation? Our in vitro transcription experiments showed that when in excess over RNAP, *Msm* HelD has an inhibitory role. Consistently, in the Hurst-Hess et al. paper[13], HelD also displayed a dampening effect on transcription in vitro. At the same time HelD also protects RNAP against rifampicin. These two effects have about the same magnitude in vitro. In the cell, however, RNAP typically associates at sub-stochiometric ratios with HelD[51]. Importantly, recent publications demonstrated that class II HelD proteins from *Msm*, *Streptomyces venezuelae*, and *Mycobacterium abscessus* protect the cells against rifampicin in transcription[13,14]. Hence, the complexity of the living system might not be fully reflected in the in vitro system.

Rifampicin binds to the β subunit of RNAP in the DNA/RNA channel, blocking transcription to proceed beyond 2–3 nt and stalling the RNAP in early initiation complex[52]. Based on available HelD–RNAP binary complex structures, a target protection mechanism of RNAP by HelD was proposed[15]. In State II & III complexes, the HelD PCh-loop folding in the AS disfavors rifampicin binding (Fig. 6b, c) and displaces the jammed DNA/RNA hybrid[9,15]. Here we bring further mechanistic details of the target protection mechanism. The us-fork–HelD–RPc structures show that HelD can dislodge rifampicin and DNA/RNA hybrid from the stalled early initiation RNAP complex while $\sigma^A$ and RbpA remain on RNAP and keep contact with the DNA promoter outside RNAP. This process would correspond to backward transition from stalled early initiation transcription complex to us-fork–HelD–RPc-II (gray dashed line arrow toward Fig. 6b). In other words, the whole early initiation assembly reverses to a closed complex-like formation, without the need for complete disassembly to individual components. The retained interactions of all the components needed for transcription initiation allow for rapid restarting of the whole initiation process. We hypothesize that the rapid restarting can reiterate until it overcomes the rifampicin inhibition. HelD appears to play a crucial role in this process by coupling the disassembly of rifampicin-stalled early initiation complexes to the new round of transcription initiation. The involvement of HelD in this process thus represents a HelD-protected mechanism of transcription initiation.

## Methods

### Strain construction

All the strains are listed in Table 1 and oligonucleotides used for strain construction are in Table 2. The concentrations of antibiotics used for selecting *E. coli* strains were as follows: ampicillin (100 μg/ml), chloramphenicol (30 μg/ml), and kanamycin (50 μg/ml). For antibiotic selection of *Msm* strains—hygromycin (50 μg/ml) and/or kanamycin (20 μg/ml) were used.

## Table 1 | Bacterial strains and plasmids

| Strain | Name | Description | Source |
|---|---|---|---|
| | [a]*E. coli* | | |
| LK_13 | DH5α | cloning strain | Laboratory strain |
| LK_222 | DH5α-pUC18 | pUC18, amp[R] | Laboratory strain |
| LK_625 | BL21 DE3 | expression strain | Laboratory strain |
| LK_2678 | Lemo21 (DE3) | expression strain, cm[R] | Laboratory strain |
| LK_1548 | P*rrnAPCL1* promoter | p770/P*rrnAPCL1*, DH5α, amp[R] | [33] |
| LK_1740 | [a]*Msm* σ[A] | pET22b/σ[A]-6xHis, DH5α, amp[R] | [9] |
| LK_1853 | *Msm* RNAP | pRMS4/*Msm* RNAP α, ω,β,β'-His(8x), BL21 DE3, kan[R] | [33] |
| LK_2647 | *Msm* HelD-FLAG | pUC18/HelD (MSMEG_2174)-FLAG, DH5α, amp[R] | This study |
| LK_2831 | TEV His(6x) | pRK793 (TEV_RIL) Lemo21 (DE3), amp[R], cm[R] | [58] |
| LK_2832 | *Msm* σ[A] | pET28bMBP/His-MBP-TEV cleavage site-σ[A] (MSMEG_2758), Lemo21 (DE3), kan[R] | This study |
| LK_2844 | *Msm* σ[A] | pET28bMBP/His-MBP-TEV cleavage site-σ[A] (MSMEG_2758), DH5α, kan[R] | This study |
| LK_2981 | *Msm* HelD | pET302/NT-His/His-TEV cleavage site-HelD (MSMEG_2174) Lemo21 (DE3), amp[R], cm[R] | [9] |
| LK_3210 | *Msm* RbpA | pET302/His-TEV cleavage site-RbpA (MSMEG_3858), BL21 DE3, amp[R] | This study |
| LK_3890 | *Msm* RbpA | pET302/His-TEV cleavage site-RbpA (MSMEG_3858), DH5α, amp[R] | This study |
| LK_4162 | HelD[σA-INT] | pET302/NT-His/His-TEV cleavage site- HelD mutant of σ[A] interface (MSMEG_2174 - D310A, W322A, Y347D, W361 A, H368A) Lemo21 (DE3), amp[R], cm[R] | This study |
| LK_4163 | HelD[A-HYDRO] | pET302/NT-His/His-TEV cleavage site- HelD mutant of ATP hydrolysis (MSMEG_2174 - E529S, Q558N) Lemo21 (DE3), amp[R], cm[R] | This study |
| LK_4164 | HelD[A-BIND] | pET302/NT-His/His-TEV cleavage site- HelD mutant of ATP binding (MSMEG_2174 – T206E) Lemo21 (DE3), amp[R], cm[R] | This study |
| LK_4165 | expression vector | pET28-MBP-TEV, DH5α, kan[R] | [54] |
| | *Msm* | | |
| LK_1321 | mc[2] 155, Recombineering vector | pJV53, kan[R] | [57] |
| LK_1468 | mc[2] 155 / RNAP-FLAG | pTE-mcs/ FLAG-DAS tag on β subunit of RNAP (MSMEG_1367), hyg[R], kan[R] | [9] |
| LK_2651 | mc[2] 155 / HelD-FLAG | C-terminal FLAG tag on HelD (MSMEG_2174), hyg[R] | This study |
| LK_2980 | mc[2] 155 | *Msm* wt | Laboratory strain |

[a]*E. coli Escherichia coli, Msm Mycobacterium smegmatis.*
All *Msm* strains are derivatives of the mc[2] 155 strain.

### Construction of *E. coli* strains for overexpression of RbpA

The gene construct coding for the N-terminally His-tagged (cleavable by TEV protease) *Msm* RbpA protein encodes from the N-terminus the MHHHHHHVNSLEENLYFQG amino acid sequence, which is followed by the MSMEG_3858 (encoding RbpA) gene starting from its second amino acid. The respective DNA fragment was prepared by PCR using Q5® High-Fidelity DNA Polymerase (NEB) with primers 3771, 3772 (MSMEG_3858, RbpA) and *Msm* mc[2] 155 chromosomal DNA (LK_2980) as the template. The PCR product was cloned into the Champion™ pET302/NT-His expression vector by the method of Restriction Free PCR cloning[53] using Phusion® High-Fidelity DNA Polymerase. The PCR reaction was treated with the restriction enzyme *DpnI* (37 °C, 3 h) and subsequently transformed into *E. coli* DH5α cells (LK_13). The resulting construct (LK_3890) was verified by sequencing. This plasmid was then transformed into *E. coli* DE3 expression cells resulting in expression strain LK_3210.

### Construction of *E. coli* strains for overexpression of σ[A]

The gene coding for the *Msm* protein σ[A] was cloned into the pET28−MBP−TEV vector (a gift from Zita Balklava & Thomas Wassmer; Addgene plasmid #69929; http://n2t.net/addgene:69929)[54] by the method of Restriction Free PCR cloning[53]. Briefly, the gene encoding σ[A] (MSMEG_2758, *mysA*) was amplified by PCR using primers TK1, TK2 (MSMEG_2758, *mysA*) from plasmid pET22b containing *mysA* (LK_1740) used as the template. The cleavage site for TEV protease was placed at the 5′ end of the gene construct. Amplified gene for σ[A] with 5′over-laping regions from the desired insertion sites at pET28−MBP−TEV was used as a primer for the second PCR reaction. After the parental plasmid elimination by *DpnI* the PCR product was transformed into

*E. coli* DH5α. The resulting protein fusion of MBP−σ[A] thus has a 6xHis tag at the N-terminus. The resulting construct (LK_2844) was verified by sequencing and then transformed into *E. coli* Lemo21 (DE3) cells (NEB) resulting in expression strains LK_2832 (σ[A]).

Preparation of *E. coli* strains for overexpression of *Msm* RNAP[33] (LK_1853) was done as described previously. Briefly, the pAC22 vector was used as a backbone vector where original genes encoding the *M. bovis* RNAP subunits were replaced with genes encoding *Msm* RNAP subunits. The *Msm rpoA* gene was inserted into the pAC22 using *XbaI* and *PacI* restriction sites. Following this, the *Msm rpoC* gene that contains an 8×His tag at the 3′ end, was inserted into the pAC22 vector via *BamHI* and *AscI* restriction sites. The *rpoB* gene, along with the sequence coding 9-amino-acid polylinker joining β and β′ subunits, was inserted using *NotI* and *AscI* restriction sites which is in-frame with *rpoC*. Finally, the *Msm rpoZ* gene was inserted into the pAC22 vector using *PacI* and *NotI* restriction sites. The resulting vector, pRMS4, encodes a polycistronic transcript that enables the expression of all five RNAP core subunits (LK_1853).

### Construction of *E. coli* strains for overexpression of HelD wild type and mutants

For production of WT *Msm* HelD, a previously described construct was used[9]. Briefly, the gene for *Msm* HelD (MSMEG_2174) with TEV clea-vable N-terminal 6xHis-tag at the 5′ end of the gene was synthetized by GeneArt® (Thermo Fisher Scientific) and then cloned into the Champion™ pET302/NT-His expression vector (Thermo Fisher Scientific) via *EcoRI* and *XhoI* restriction sites (name of the expression strain: LK_2981, Table 1). Mutated variants of HelD (LK_4162, LK_4163, LK_4164) were prepared by site-directed mutagenesis using PCR (using

**Table 2 | Oligonucleotides used in the study'**

| Oligonucleotides | Sequence 5 → 3' | Description |
|---|---|---|
| TK1 | CAGGAGAACCTGTACTTCCAGGGCATGGCAGCGACAAAGGCAAG | Primers for cloning of the *Msm* σ$^A$ gene (MSMEG_2758) into pET28-MBP-TEV |
| TK2 | TGCCCTGGAAGTACAGGTTTTCTTCTCGAGCTAGTCCAGGTAGTCGC | |
| TK71 | GCTTGACAAAAGTGTTAAATTGTGCTATACT | [20] |
| TK72 | CAATTTAACACTTTTGTCAAGC | this study |
| 3286/ HelD-FLAG_ra_F | AACTTCTCTAGACTCAGTTCGCACACGCCTG | Primers for the right homology arm of HelD (MSMEG_2174) |
| 3287/ HelD-FLAG_ra_R | TACGAATTCGAGCTCGGTACCCGGGGATCCCTCGACCCGTTCGTCGAC | |
| 3290/ HelD-FLAG_la_F | CGTTGTAAACGACGGCCAGTGCCAAGCTTCGTGGGAGACCTGGCGCAG | Primers for the left homology arm of HelD (MSMEG_2174) |
| 3291/ HelD-FLAG_la_R | CGTTAACCTGCAGCTACTTGTCGTCGTCGTCCTTGTAG | |
| 3292/ hygro-HelD-FLAG_F | GACGACAAGTAGCTGCAGGTTAACGAAATCAATC | Primers for the hygromycin resistance |
| 3293/ hygro-HelD-FLAG_R | TGTGCGAACTGAGTCTAGAGAAGTTATCCCGGG | |
| 3771/ RbpA_No_tag_F | CGAGGAAAACCTGTACTTCCAGGGTATGGCTGATCGTGTCCTGCGG | Primers for cloning of *Msm rbpA* (MSMEG_3858) into pET302/NT-His. |
| 3772/ RbpA_No_tag_R | CTTTCGGGCTTTGTTAGCAGCCGGATCCTTAGCTTCCGGTTCCGCGCCG | |
| 4394/nt-strand_OC | GCTTGACAAAAGTGTTAAATTGTGCTATACTGGGAGCCGTCACGGATGCG | Non-template strand to form open double-stranded DNA resembling transcription bubble[30]. |
| 4395/t-strand_OC | CGCATCCGTGAGTCGAGGGTAATAAGCACAATTTAACACTTTTGTCAAGC | Template strand to form open double-stranded DNA resembling transcription bubble |
| 4396/nt-strand_CC | GCTTGACAAAAGTGTTAAATTGTGCTATACT | Non-template strand DNA to form closed double-stranded DNA resembling the closed promoter complex[30] |
| 4397/t-strand_CC | AGCACAATTTAACACTTTTGTCAAGC | Template strand DNA to form closed double-stranded DNA resembling the closed promoter complex |
| JD1/HelD sigA-INT_F | TCGACCTGTCTGCGGTTACCATGCGTATCGACGCTGAAACCGCTAAAGCGGCTCGTGACGAAGCT | Forward primer for mutagenesis of HelD to create LK_4162 (mutant of σ$^A$ interaction interface) (MSMEG_2174 - D310A, W322 A, Y347D, W361A, H368A) |
| JD2/HelD sigA-INT_R | CATTTTTTCCCAAGCGGCTTTGTCGTCACGGGTCAGCGCACCACGACCGATACGAGCATCAGCACGTTCGGTAACAACATCGGTAACAACGTCAAC | Reverse primer for mutagenesis of HelD to create LK_4162 (mutant of σ$^A$ interaction interface) (MSMEG_2174 - D310A, W322A, Y347D, W361A, H368A) |
| JD3/HelD sigA-HYDRO_F | CGTTGTTGTTGACAGCGCTCAGGAACTGTCTGAAAT | Forward primer for mutagenesis of HelD to create LK_4163 (mutant - ATP hydrolysis) (MSMEG_2174 - E529S, Q558N) |
| JD4/HelD sigA-HYDRO_R | CGGAGAACGACGGTTAGCCAGGTCGCCAA | Reverse primer for mutagenesis of HelD to create LK_4163 (mutant - ATP hydrolysis) (MSMEG_2174 - E529S, Q558N) |
| JD5/HelD sigA-BIND_F | TCCGGGTACCGGTAAAGAAGTTGTTGCTCTGCA | Primer for mutagenesis of HelD to create LK_4164 (mutant - ATP binding) (MSMEG_2174 – T206E) |

Q5® High-Fidelity 2X Master Mix [NEB]) with the specific primers listed in Table 2. The construct containing the wild type form of HelD (LK_2981) described above was used as a template. The resulting constructs were verified by sequencing and then transformed into expression strain *E. coli* Lemo21 (DE3).

### Construction of the *Msm* strain containing HelD−FLAG
The HelD strain with a FLAG-tagged version of HelD knocked-in in the genome (native locus) was generated as follows. First, the cassette for knock-in was generated by combining pUC18 (LK_222) digested with *HindIII* and *EcoRI* with three PCR fragments: 500 bp of the left homology arm containing the FLAG tag (primers 3290, 3291), (C-terminally appended to the HelD gene [MSMEG_2174]) followed by a hygromycin resistance encoding sequence (primers 3292, 3293)[55,56] and 500 bp of the right homology arm (primers 3286, 3287). The fragments were assembled with a Gibson assembly kit (NEB). The Gibson assembly mixture was transformed into *E. coli* DH5α. The resulting strain (LK_2647) was verified by sequencing. The fragment encompassing the cassette was subsequently transformed into the *Msm* pJV53 strain (LK_1321; this strain has an increased frequency of homologous recombination[57]) and individual clones were selected for hygromycin resistance. Subsequently, the clones were cured of pJV53,

and one clone was selected, which was hygromycin resistant and kanamycin sensitive (LK_2651) and was used in further studies.

### Growth conditions
*Msm* strains used in this study were streaked out from glycerol stocks onto solid agar-based media (Middlebrook 7H10) and allowed to grow for two to three days at 37 °C. Then, they were inoculated into Middlebrook 7H9 medium with 0.2% glycerol and 0.05% Tween 80 at 37 °C and grown overnight. Next day, they were inoculated into the same medium at starting $OD_{600}$ - 0.1 and grown as specified.

*E. coli* strains used for overexpression of the proteins−RNAP (LK_1853), RbpA (LK_3890)−were grown overnight in LB media at 37 °C. On the following day, they were inoculated into the same medium at a starting of $OD_{600}$ - 0.03 and grown as specified.

### *Msm* HelD and RNAP pull down
*Msm* strains: No-Tag (LK_2980) and HelD−FLAG (LK_2651), RNAP−FLAG (LK_1468) were grown in 7H9 media using appropriate antibiotics (see Table 1). From the exponential phase of growth ($OD_{600}$ = 0.5), 150 ml of bacterial culture was harvested while 50 ml was harvested from the stationary phase (24, 48, and 72 h after inoculation as specified for individual strains). The pellet was washed and re-suspended in 3 ml of

lysis buffer (20 mM Tris-HCl, pH 8.0; 150 mM KCl; 1 mM MgCl$_2$; 0.5 mM 1,4-dithiothreitol (DTT); 0.5 mM PMSF; 5 µl/ml of protease inhibitor cocktail [Sigma P8849]), sonicated with a Hielscher UP200S ultrasonic processor (15 × 10 s on ice, amplitude 50 %, 1 min break on ice) and centrifuged to collect the cell lysate. Anti-FLAG M2 affinity gel (Sigma-A2220) conjugated with agarose was added to the 1.5 ml of lysate (adjusted to contain the same amount of protein) and incubated for 14 h to allow for antibody–antigen complex formation. To remove nucleic acids from HelD-FLAG pull-down, 1 µl (25 U) of Benzonase (Qiagen-Merck KGaA-1038893) was added to the lysate. Both benzonase-treated and untreated cell lysates were used for HelD–FLAG pull-down. The agarose beads containing FLAG-tagged proteins were then washed four times with 1 ml of lysis buffer. Next, 3xFLAG peptide elution solution in TBS buffer (Sigma-F4799, final concentration 150 ng/ml) was added to the beads. After 4 h of incubation at 4 °C, the FLAG-tagged proteins were collected by centrifugation and consequently checked by SDS-PAGE and sent for mass spectrometry (MS) analysis. Protein marker used in SDS PAGE gels shown in Figs. 1a, 4d–f, Supplementary Fig. 1a, and Supplementary Fig. 15 was Novex™ Sharp Pre-stained Protein Standard (Thermo Fisher Scientific). Protein marker used in Fig. 4g was Spectra™ Multicolor Broad Range Protein Ladder (Thermo Fisher Scientific). Protein marker used in Supplementary Fig. 2c was PageRuler™ (Thermo Fisher Scientific).

## Protein expression and purification
**Purification of *Msm* RNAP-8xHis.** The LK_1853 *E. coli* strain carrying the pRMS4 plasmid[33] encoding *Msm* RNAP was grown in LB medium in the presence of kanamycin (50 µg/ml) until it reached OD$_{600}$ of 0.6. Protein expression was then induced by 0.5 mM IPTG and the culture was incubated for 4 h at room temperature. The bacterial cells were harvested by centrifugation, washed with P buffer (300 mM NaCl; 50 mM Na$_2$HPO$_4$; 5% glycerol; 3 mM β-mercaptoethanol) and resuspended again in 10 ml of P buffer for subsequent steps. The cells were lysed by sonication (Hielscher UP200S ultrasonic processor, 12 × 10 s on ice, amplitude 50%, 1 min break) and centrifuged. To isolate RNAP, the supernatant was mixed with 1 ml of Ni-NTA Agarose beads (Qiagen) and incubated for 90 min at 4 °C with gentle shaking. The Ni-NTA Agarose with bound RNAP was loaded onto a Poly-Prep Chromatography Column (Bio-Rad). The column was washed first with 30 mL of P buffer and then with 30 mL of P buffer containing 30 mM imidazole. The bound proteins were then eluted using P buffer containing 400 mM imidazole. The fractions containing RNAP were pooled together, dialyzed into storage buffer (50 mM Tris-HCl, pH 8.0; 100 mM NaCl; 50% glycerol; 3 mM β-mercaptoethanol) and kept at −20 °C.

## Purification of tag-less *Msm* RbpA
The LK_3210 *E. coli* strain carrying the expression vector pET302 with RbpA (MSMEG_3858) was grown in LB medium in the presence of ampicillin (100 µg/ml) until it reached early exponential phase (OD$_{600}$ of 0.5). Protein expression was induced by 0.5 mM IPTG, and the culture was further grown for 4 h at room temperature. The bacterial cells were harvested, and the same protocol as for RNAP purification was followed. Fractions containing the eluted protein were collected and pooled, and subsequently dialyzed against TEV cleavage buffer. TEV protease (LK_2831) was prepared as described[58] and added to the dialyzed proteins at a TEV protease: protein ratio of 1:20 and the cleavage was allowed to proceed for 16 h at 4 °C. The cleaved protein was again dialyzed against binding buffer for affinity chromatography. The dialyzed protein was loaded onto a HisTrap HP affinity column (GE17-5247-01), and the His-TEV part was removed from the mixture, thereby eluting the pure RbpA. The peak fractions of RbpA were pooled and concentrated using an Amicon centrifugal filter (3 kDa MWCO). Finally, the purified RbpA was dialyzed against storage buffer and kept at −20 °C.

## Purification of tag-less *Msm* HelD and mutants
The LK_2981, LK_4162, LK_4163, and LK_4164 *E. coli* strains carrying the pET302 plasmid encoding *Msm* HelD[9] and its mutants (consequently, see Table 1) were first grown overnight in PB media (Molecular Dimensions, Rotherham, UK) at 37 °C in the presence of carbenicillin (50 µg/ml) and chloramphenicol (30 µg/ml). Resulting "overnight" cultures were the next day diluted at ratio 1:100 into 1 l (in 5 l flasks) of fresh Overnight Express™ Instant TB media (Novagen, Merck, Darmstadt, Germany) each supplemented with carbenicillin (25 µg/ml) and chloramphenicol (15 µg/ml), and grown at 37 °C/180 rpm till they reach OD$_{600}$ = 0.6. Then temperature was set to 20 °C and they were further grown for 16 h with the same shaking speed. Afterwards, cells were harvested by centrifugation (4000 *g*, 4 °C, 20 min) and stored at −80 °C. Bacterial pellets were resuspended in binding buffer: 50 mM Tris-HCl pH 7.5, 500 mM NaCl, 30 mM imidazole, 0.2% (v/v) Tween 20 supplemented with hen egg white lysozyme (final concentration 0.2 mg/ml, Sigma-Aldrich, Darmstadt, Germany). 0.5 mg of DNase I from bovine pancreas (Sigma-Aldrich, Darmstadt, Germany) and 0.5 ml of Protease Inhibitor Cocktail (Sigma-Aldrich, Darmstadt, Germany) per 10 g of wet weight were added and suspension was incubated on ice with stirring for 30 min. Suspension was then sonicated on ice, centrifuged for 30 min at 40,000 g and 4 °C. Supernatant was filtered through 0.22 µm filter and purified using Ni−NTA chromatography, 1 ml HisTrap™ FF column (GE Healthcare, Cytiva, Marlborough, MA, USA) and an ÄKTA Purifier (GE Healthcare). In the next step, TEV cleavage was performed. Proteins were incubated in 50 mM Tris-HCl pH 7.5, 100 mM NaCl, 0.5 mM TCEP, 0.5 mM DTT, 1 mM EDTA at 4 °C overnight. Each sample was then run through 1 ml HisTrap™ FF column (GE Healthcare, Cytiva, Marlborough, MA, USA) using an ÄKTA purifier and flow-through fractions were stored. The final purification step was size-exclusion chromatography performed using Superdex 200 Increase 10/300 GL column (GE Healthcare), 25 mM Tris-HCl pH 7.5, 100 mM NaCl, 10 mM MgCl$_2$, 0.5 mM TCEP buffer, and 0.6 ml/min flow rate.

## *Msm* HelD ATP and GTP hydrolysis assays
Hydrolysis of ATP and GTP by *Msm* HelD and its complexes with *Msm* RNAP + σ$^A$+RbpA or *Msm* RNAP core, by RbpA, and by σ$^A$ were measured as follows. The complexes were purified before measurement using SEC (as in the previous paragraph) and the amount of complex used in hydrolysis fulfilled the following condition: 50 µl of reaction mixture contained 5 mM substrate (ATP or GTP), 5 µg of *Msm* HelD or the same amount of *Msm* HelD in an equimolar ratio complex with RNAP + σ$^A$+RbpA or RNAP core. Separately, hydrolysis activities of the same amount of RNAP holoenzyme, RNAP core, σ$^A$ and RbpA were tested as well. 25 mM Tris-HCl pH 7.5, 100 mM NaCl, 10 mM MgCl$_2$, 0.5 mM TCEP, was used as the reaction buffer. The reaction mixtures were incubated at 37 °C for 90 min.

The ATP and GTP hydrolysis activities were analyzed spectrophotometrically at $\lambda$ = 850 nm by monitoring the amount of released phosphate according to a modified molybdenum blue method[59] using a microplate reader Clariostar (BMG LABTECH, Ortenberg, Germany). Briefly, the reactions were stopped by adding 62 µl of the solution A containing 0.1 M L-ascorbic acid, 0.5 M trichloroacetic acid. After thorough mixing, 12.5 µl of reagent B (10 mM ammonium molybdate) and 32 µl of reagent C (0.1 M sodium citrate, 0.2 M sodium arsenite, 10% acetic acid) was added. All enzymatic reactions were performed in triplicates with separate background readings for each condition. The amount of released phosphate in enzymatic reactions was determined using calibration curve. The data were analyzed using GraphPad Prism 7.02 (GraphPad Software, San Diego, California USA, www.graphpad.com).

## HeD release assay
RNAP−protein complexes were reconstituted in 10 µl of transcription buffer (40 mM Tris-HCl, pH 8.0; 10 mM MgCl$_2$; 1 mM DTT). Each

reconstitution contained 10 pmol of RNAP–His and/or 50 pmol of $\sigma^A$ and/or 30 pmol of RbpA, and 30 pmol of HelD (WT/Mutants). Reconstitution was carried out in 1.5 ml eppendorf tubes for 5 min at 25 °C with 260 rpm shaking in a thermostatic shaker (TS100C Biosan). Cobalt-coated magnetic beads (Dynabeads, Invitrogen-10103D) with affinity for poly-histidine tags (5 µl of beads per reaction) were washed with transcription buffer and then resuspended in transcription buffer (5 µl per reaction). 0.2 mg (5 µl) of the washed beads was added to 10 µl of reconstituted proteins (one reaction). The protein-bead mixture was incubated for 10 min at 25 °C with 260 rpm shaking. All unbound proteins were removed by placing the protein-bead mixture in a DynaMag magnet (Invitrogen- 12321D) and washed with 100 µl of transcription buffer according to the manufacturer's protocol.

To check the amount of HelD bound to RNAP (association of the complexes), the beads with the bound RNAP complex were resuspended in 10 µl of transcription buffer and heated to 95 °C for 5 min. The beads were then separated using a magnet, and the supernatant containing the released proteins after heat treatment was analyzed on SDS-PAGE gel and scanned with Epson V850 Photo Scanner. The individual protein bands were quantified with Quantity One software (Bio-Rad). The protein band intensities were plotted with ggplot2[60] in R (R Core Team (2022). R: A language and environment for statistical computing. R Foundation for Statistical Computing, Vienna, Austria. https://www.R-project.org/) with default settings (whiskers extend to the highest (lowest) value within 1.5×IQR beyond upper (lower) quartile). For representative examples see Supplementary Fig. 18.

For NTP/DNA induced HelD release assays, the beads with bound RNAP complex were resuspended in 5 µl/reaction of transcription buffer. This mixture was then combined with 5 µl containing 1 µl of 10×R buffer (200 mM Tris-HCl, pH 8.0; 1 M KCl; 11 mM MgCl$_2$) and the following compounds or their combinations that were used to induce HelD release. The compounds were: DNA (OC/CC, see the next paragraph for details), 1 mM ATP, 1 mM GTP, 1 mM CTP, 1 mM non-hydrolyzable ATP (adenosine 5′-(β,γ-imido)triphosphate [Merck-A2647]), and 1 mM ATPγS (adenosine 5′-O-(3-thio)triphosphate [Merck-11162306001]). These compounds were added to the reactions as specified in individual experiments. The reaction was allowed to proceed for 10 min at 25 °C with 260 rpm in the thermo shaker. Then, the beads were separated using a magnet, and the supernatant containing the released HelD was collected. The HelD released from each individual reaction was analyzed on SDS-PAGE gel and quantified as mentioned above. The experiments were conducted in at least three biological replicates.

To form closed and open transcription complexes with DNA promoters, oligonucleotides were purchased (4394, 4395 for open complex−OC; 4396, 4397 for closed complex−CC, see Table 2). In total, 15 µl of each oligonucleotide (100 pmol/µl) was used to obtain the complexes. The annealing was done in a thermocycler (98 °C for 5 min, 95 °C for 1 min, the temperature was decreasing by 1 °C every 1 min, 70 cycles in total). In total, 50 pmol of the reconstituted DNA (OC/CC) was used in individual reactions.

As a control for the HelD release experiments to ascertain that we reliably detect the relative amounts of released HelD, we used a calibration curve of known HelD concentrations and verified that the signal of released HelD was within the linear part of the calibration curve. The release experiment and calibration curve control were done in parallel. Increasing amounts of HelD (0.5, 1, 2, 4 pmol) were prepared in 10 µl of transcription buffer and analyzed on SDS-PAGE. The densitometric volume for each protein band was quantified with Quantity One software (Bio-Rad) and a calibration curve was obtained. In parallel, NTP induced HelD release assay was performed as specified above. HelD released from the reaction was analyzed on SDS-PAGE and quantified. The band intensities of released HelD for individual reactions were plotted on the calibration curve (for an example see Supplementary Fig. 14). Data for Fig. 4a, b, d–g were plotted using SigmaPlot (version 8.0).

## Multiple round in vitro transcriptions

Multiple round in vitro transcription assays were carried out as described previously[61,62] unless stated otherwise. See the following paragraphs for a detailed description. $\sigma^A$-dependent ribosomal RNA promoter P*rrnAPCL1* from *Msm* (LK_1548) was used[33] in the transcription reactions. All in vitro transcription reactions were stopped by addition of 10 µl of formamide stop solution (95% formamide, 20 mM EDTA, pH 8.0, 0.03% bromophenol blue, 0.03% xylene cyanol FF)[63]. Samples were desaturated for 5 min at 95 °C and loaded on polyacrylamide (PAA) gel (7% PAA, 7 M urea). Gels were run for 120 min at 170 V. Gels were dried for 1 h at 80 °C, cooled down and exposed overnight on BAS storage phosphor screen (Fujifilm). Subsequently, the screen was scanned using Amersham™ Typhoon™ 5 Biomolecular Imager (Cytiva) with a phosphor imaging emission filter 390BP. The signal was quantified with the QuantityOne (Bio-Rad, version 4.6.3) software and plotted using SigmaPlot (version 8.0). Statistical calculations were done in Microsoft Excel (Office 365, version 23–24).

For in vitro transcriptions with rifampicin (Fig. 5a), first RNAP (LK_1853), RbpA (LK_3210), and $\sigma^A$ (LK_2832) were reconstituted in storage buffer (0.3 µM RNAP, 1.5 µM RbpA, and 6 µM $\sigma^A$) in a final volume of 10 µl. These proteins were then incubated for 10 min at 37 °C. Following this incubation, 25 ng (final concentration 7.32 nM) of supercoiled plasmid DNA (LK_1548) per 10 µl reaction was added to the reconstituted proteins and incubated for another 10 min at 37 °C. Subsequently, rifampicin was added at final concentrations of 12.5 nM, 50 nM, 200 nM, or no rifampicin (EREMFAT, Riemser Arzneimittel, diluted in DMSO), and the reaction mixture was again incubated for 10 min at 37 °C. Finally, HelD (LK_2981) was either added or not, at a final concentration of 0.3 µM in the final 10 µl reaction volume, followed by another 10 min incubation at 37 °C (tube A). A separate reaction mixture (7 µl per reaction) was prepared containing transcription buffer (40 mM Tris-HCl, pH 8.0; 10 mM MgCl$_2$; 1 mM DTT, 0.1 mg/ml BSA, 50 mM KCl, and NTPs (200 µM ATP and CTP; 5 mM GTP; 10 µM UTP; 2 µM of radiolabeled [α$^{32}$P]-UTP [Hartmann Analytic]). This mixture (tube B) was then allowed to equilibrate at 37 °C for 5 min. Transcriptions were then initiated by adding 3 µl of the reconstituted proteins and DNA template ±rifampicin (from tube A) to 7 µl of tube B, and allowed to proceed for 10 min at 37 °C.

For in vitro transcriptions with increasing amounts of HelD and RbpA (Supplementary Fig. 19a, b), reactions were carried out in 10 µl: 25 ng of supercoiled DNA template (final concentration 7.32 nM, LK_1548), transcription buffer (40 mM Tris-HCl, pH 8.0; 10 mM MgCl$_2$; 1 mM DTT, 0.1 mg/ml BSA, 50 mM KCl and NTPs (5 mM GTP, 200 µM ATP and CTP; 10 µM UTP; 2 µM of radiolabeled [α$^{32}$P]-UTP). Transcriptions were initiated with 2 µl of reconstituted proteins (RNAP + $\sigma^A$ ± RbpA ±HelD) yielding a final volume of 10 µl. The final concentrations of the reconstituted proteins were: RNAP (LK1853), 0.5 µM; $\sigma^A$ (LK_2832), 10 µM; RbpA (LK_3210), 0.5 µM, 2.5 µM, 20 µM; HelD (LK_2981), 0.5 µM, 2 µM, 4 µM. Protein reconstitutions were carried out for 10 min at 37 °C. In vitro transcriptions were allowed to proceed for 10 min at 37 °C.

For in vitro transcriptions with increasing amounts of GTP in presence or absence of HelD (Supplementary Fig. 20a), reactions were set up as described in the previous paragraph. The only difference was that GTP concentrations were titrated: 20 µM, 40 µM, 100 µM, 200 µM, 400 µM, 600 µM, 1000 µM, 1500 µM, 2000 µM, 3500 µM, and 5000 µM. Transcriptions were initiated with 2 µl of reconstituted proteins (RNAP + RbpA + $\sigma^A$ ± HelD) yielding a final volume of 10 µl. The final concentrations of the reconstituted proteins were: RNAP (LK_1853), 0.3 µM; RbpA (LK_3210), 1.5 µM; $\sigma^A$ (LK_2832), 6 µM; HelD (LK_2981), 1.2 µM. Protein reconstitutions were carried out for 10 min at 37 °C. In vitro transcriptions were allowed to proceed for 10 min at 37 °C. After signal was quantified as described above, the exponential rise to maximum function of Sigmaplot was used to fit the data. $K_{NTP}$

values were calculated from the $f = a*[1 − \exp(−b*x)]$ equation ($f$ = relative transcription; $x$ = time; $a$ and $b$ = constants)[42].

## LC−MS/MS analysis

**Sample preparation.** Proteins were digested with 0.1 μg of trypsin solution in 50 mM ammonium bicarbonate at 37 °C for 16 h. The resulting peptides were separated on an UltiMate 3000 RSLCnano system (Thermo Fisher Scientific) coupled to an Orbitrap Fusion Lumos mass spectrometer (Thermo Fisher Scientific). The peptides were trapped and desalted with 2% acetonitrile in 0.1% formic acid at a flow rate of 30 μl/min on an Acclaim PepMap100 column [5 μm, 5 mm by 300-μm internal diameter (ID); Thermo Fisher Scientific]. The eluted peptides were separated using an Acclaim PepMap100 analytical column (2 μm, 50 cm by 75 μm ID, Thermo Fisher Scientific). The 125-min elution gradient at a constant flow rate of 300 nl/min was set to 5% of phase B (0.1% of formic acid in 99.9% of acetonitrile) and 95% of phase A (0.1% of formic acid) for 1 min, after which the content of acetonitrile was gradually increased. The Orbitrap mass range was set from m/z 350 to 2000 in the MS mode, and the instrument in Data dependent acquisition (DDA) mode acquired HCD fragmentation spectra for ions of m/z 100–2000.

## Protein identification and quantification

MaxQuant with Andromeda search engine (version 1.6.3.4; Max−Planck-Institute of Biochemistry, Planegg, Germany) was utilized for peptide and protein identification and identification with databases of the *Msm* proteome (downloaded from UniProt on 20th of December 2019) and common contaminants. Following settings were used: Fixed modification: carbamidomethyl (C); variable modifications: oxidation (M), acetyl (protein N-term); enzyme: Trypsin/P, 2 missed cleavages allowed; match between runs was enabled. Mass tolerance for the peptide first and main search was set as 20 ppm and 4.5 ppm, respectively. Minimal peptide length was set as 7. FDR level was set as 0.01 for both peptides and proteins. The dataset of samples treated with benzonase was searched together with dataset without benzonase. The dataset with benzonase was used for Fig. 1 as it contained fewer potential background proteins that could have been pulled down via nucleic acids associated with RNAP. The PRIDE submission contains all data. Perseus software (version 1.6.2.3; Max−Planck-Institute of Biochemistry) was used for the label-free quantification of HelD-FLAG after benzonase treatment compared to negative control (wild type strain without FLAG tag) at two different time points (exponential and stationary growth phase). The identified proteins were filtered for contaminants and reverse hits. Proteins detected in the data were filtered to be quantified in at least two of the triplicates. The data were processed to compare the abundance of individual proteins by statistical tests in the form of student's *t*-test and resulted in a volcano plot comparing the statistical significance (two-sided p-value) and protein-abundance difference (fold change). The statistical test was performed with three of four biological replicates, due to the significantly lower number of identified proteins in one of the replicates. The mass spectrometry proteomics data have been deposited to the ProteomeXchange Consortium via the PRIDE[64] partner repository with the dataset identifier PXD046632 and doi:10.6019/PXD046632.

## Electron microscopy

### In vitro complex reconstitution for cryo-EM

**RNAP−HelD−σA−RbpA complex.** *Msm* RNAP core, HelD, σA and RbpA proteins for in vitro reconstitution of cryo-EM samples were purified as described previously[9]:

***Msm* RNAP core purification for cryo-EM.** *E. coli* strain BL21(DE3) was transformed with pRMS4 (kanR) plasmid derivative encoding *Msm* subunits ω, α, and β–β′ fusion with C-terminal His8 tag in one operon from the T7 promoter[9]. Expression cultures were incubated at 37 °C

and shaken at 250 rpm until OD600 ~ 0.8; expression was induced with 500 μM isopropyl β-d-thiogalactoside (IPTG) at 17 °C for 16 h. Cells were lysed using sonication by Sonic Dismembrator Model 705 (Thermo Fisher Scientific) in a lysis buffer containing 50 mM NaH$_2$PO$_4$/Na$_2$HPO$_4$ pH 8 (4 °C), 300 mM NaCl, 2.5 mM MgCl$_2$, 30 mM imidazole, 5 mM β-mercaptoethanol, EDTA-free protease inhibitor cocktail (Roche), RNase A (Sigma), DNase I (Sigma), and Lysozyme (Sigma). Clarified lysate was loaded onto a HisTrap FF Crude column (Cytiva) and proteins were eluted with a linear gradient of imidazole to the final concentration of 400 mM over 20 column volumes. The *Msm* RNAP core elution fractions were pooled and dialyzed to 20 mM Tris-HCl pH 8 (4 °C), 1 M NaCl, 5% (v/v) glycerol and 4 mM dithiothreitol (DTT) for 20 h. The protein was further polished on XK 26/70 Superose 6 pg column (GE Healthcare) equilibrated in 20 mM Tris-HCl pH 8 (4 °C), 300 mM NaCl, 5% (v/v) glycerol, and 4 mM DTT. The *Msm* RNAP core final fractions were eluted at 6 μM concentration, flash-frozen in liquid nitrogen, and stored at −80 °C.

***Msm* HelD purification for cryo-EM.** *E. coli* strain Lemo 21 (DE3) was transformed with pET302/NT-His (cmlR and ampR) plasmid derivative encoding the *Msm* HelD protein fusion with N-terminal 6×His tag under the control of the T7 promoter[9]. Expression cultures were incubated at 37 °C and shaken at 250 rpm until OD600 ~0.8; expression was induced with 500 μM IPTG at 17 °C for 16 h. Cells were lysed using sonication by Sonic Dismembrator Model 705 (Thermo Fisher Scientific) in a lysis buffer containing 50 mM Tris-HCl pH 7.5 (4 °C), 400 mM NaCl, 30 mM imidazole, 0.2% Tween20, 2 mM β-mercaptoethanol, EDTA-free protease inhibitor cocktail (Roche), RNase A (Sigma), DNase I (Sigma), and Lysozyme (Sigma). Clarified lysate was loaded onto a HisTrap FF Crude column (Cytiva) and proteins were eluted with a linear gradient of imidazole to the final concentration of 400 mM over 20 column volumes. Fractions containing HelD protein were pooled and dialyzed for 20 h against the dialysis buffer containing 20 mM Tris-HCl, pH 7.5 (4 °C), 500 mM NaCl, 1 mM DTT together with TEV protease at a TEV protease:HelD ratio 1:20. The protein was then concentrated to ~15 A280 units and further purified using size-exclusion chromatography using a Superdex 75 column (Cytiva) equilibrated in 20 mM Tris-HCl, pH 7.5 (4 °C), 200 mM NaCl and 1 mM DTT. The HelD protein was eluted at ~160 μM concentration and stored at −80 °C.

***Msm* σA purification for cryo-EM.** Expression strain of *E. coli* containing plasmid with gene of σA (LK1740) was grown at 37 °C until OD$_{600}$ reached ~0.5; expression of σA was induced with 300 μM IPTG at room temperature for 3 h. Isolation of σA was done in the same way as *Msm* RNAP-8×His purification (see Protein expression and purification section) with the exception of 50 mM imidazole added to the P buffer before resuspending the cells. Instead of the purification in a column, batch purification and centrifugation were used to separate the matrix and the eluate.

***Msm* RbpA purification for cryo-EM.** The expression and purification of RbpA (LK1254, this work) were done in the same way as for *Msm* RNAP-8xHis (see Protein expression and purification section) except when OD$_{600}$ reached ~0.5, the expression was induced with 800 μM IPTG at room temperature for 3 h.

To assemble the RNAP−HelD−σA−RbpA complex, the individual proteins were mixed at a molar ratio of 1:3:3:5, respectively. The in vitro reconstitutions were carried out at 4 °C, and the reconstitution mixture was incubated for 15 min. 50 μl of the reconstitution mixture was injected onto a Superose 6 Increase 3.2/300 column (GE Healthcare) equilibrated in 20 mM Tris-HCl, pH 7.8 (4 °C), 150 mM NaCl, 10 mM MgCl$_2$ and 1 mM DTT. 50 μl fractions were collected and the protein was eluted at ~1.25 μM concentration. Absorbance of the elution fraction at A$_{280}$ and A$_{254}$ was measured on the fly by UV monitor U9-M of an ÄKTA pure (Cytiva).

## RNAP–HelD–σ$^A$–RbpA complex with upstream fork DNA

The DNA fragments TK71 and TK72 were annealed at equimolar ratio. RNAP–HelD–σ$^A$–RbpA–TK71/72 were then mixed at a molar ratio of 1:3:3:5:5, respectively. The in vitro reconstitutions were carried out at 4 °C, and the reconstitution mixture was incubated for 15 min. Fifty microlitres of the reconstitution mixture was injected onto a Superose 6 Increase 3.2/300 column (GE Healthcare) equilibrated in 20 mM Tris-HCl, pH 7.8 (4 °C), 150 mM NaCl, 10 mM MgCl$_2$ and 1 mM DTT. 50 µl fractions were collected and the protein was eluted at ~1.0 µM concentration. Absorbance of the elution fraction at 280 and 254 nm was measured on the fly by UV monitor U9-M of an ÄKTA pure (Cytiva).

## RNAP–HelD–σ$^A$–RbpA complex with transcription bubble DNA

The DNA fragments 4394/nt-strand_OC and 4395/t-strand_OC were annealed at equimolar ratio to form the transcription bubble DNA. RNAP–HelD–σ$^A$–RbpA complex was reconstituted from individual proteins mixed at a molar ratio of 1:2.5:2.5:5, respectively, and SEC purified on Superose 6 Increase 3.2/300 column (GE Healthcare) equilibrated in 20 mM Tris-HCl, pH 7.8 (4 °C), 150 mM NaCl, 10 mM MgCl$_2$ and 1 mM DTT. 30 µl fraction of RNAP–HelD–σ$^A$–RbpA complex at 0.9 µM concentration was mixed with 1.2 molar excess of the transcription bubble DNA and incubated 15 min at 4 °C.

## Electron microscopy data collection

Aliquots of 3 µl, at ~0.8–1.2 µM, were applied to Quantifoil R1.2/1.3 or R2/1 Au 300 mesh grids, immediately blotted for 2 s and plunged into liquid ethane using a FEI Vitrobot IV (4 °C, 100% humidity).

The grids were loaded into an FEI Titan Krios electron microscope at the European Synchrotron Radiation Facility (ESRF) (beamline CM01, ESRF) or CEITEC (Masaryk University, Brno), operated at an accelerating voltage of 300 keV and equipped with a post-GIF K2 Summit or K3 BioQuantum direct electron camera (Gatan) operated in counting mode. Cryo-EM data was acquired using EPU software (FEI) or SerialEM[65]. Data collection details are listed in (Supplementary Table 4).

## Cryo-EM image processing

All movie frames were aligned using MotionCor2[66]. Thon rings from summed power spectra of every 4e$^-$/Å$^2$ were used for contrast transfer function parameter calculation with CTFFIND 4.1[67]. Particles were selected with TOPAZ[68] using trained picking models for each individual dataset. Further 2D and 3D cryo-EM image processing was performed in RELION 4.0[69,70] as illustrated in cryo-EM supplementary information (Supplementary Figs. 3, 4, 6–9). The final cryo-EM density maps were generated by the post-processing feature in RELION and sharpened or blurred into MTZ format using CCP-EM[71]. The resolutions of the cryo-EM density maps were estimated at the 0.143 gold standard Fourier Shell Correlation (FSC) cut off. A local resolution was calculated using RELION and reference-based local amplitude scaling was performed by LocScale[72]. The directional resolution anisotropy was quantified by the 3D FSC algorithm version 3.0[73]. The angular orientation distribution of the 3D reconstruction was calculated by cryoEF v1.1.0[74].

## Cryo-EM model building and refinement

Atomic models of *Msm* protein parts were generated according to the known structures of the *Msm* HelD–RNAP complex (PDB entry: 6YXU[9] and 6YYS[9]), *Msm* transcription initiation complex and open complex (PDB entry: 5TW1[20] and 5VI8[30]). The whole assemblies were first rigid-body fitted into the cryo-EM density by Molrep[75] and individual subdomains fits were optimized using the Jigglefit tool[76] in Coot[76,77] and best fits were chosen according to a correlation coefficient in the JiggleFit tool. Fit of individual domains was manually edited and the rest

of the proteins were built *de-novo* in Coot. The cryo-EM atomic-models of the target complexes were then iteratively improved by manual building in Coot, using ISOLDE[78] in UCSF ChimeraX[79] and refinement and validation using Phenix real-space refinement[80,81]. DNA conformation was validated using DNATCO[82]. The atomic models were validated with the Phenix validation tool (Supplementary Table 4) and the model resolution was estimated at the 0.5 FSC cut-off. Structures were analyzed and figures were prepared using the following software packages: PyMOL (Schrödinger, Inc.), USCF Chimera[83], CCP4MG[84], ePISA server[85].

## Reporting summary

Further information on research design is available in the Nature Portfolio Reporting Summary linked to this article.

## Data availability

Atomic models coordinates and cryo-EM maps have been deposited in wwPDB and EMDB: *Msm* HelD–σ$^A$–RbpA–RNAP complex State I: EMD-18128, PDB ID 8Q3I, *Msm* HelD–σ$^A$–RbpA–RNAP complex State II: EMD-18511, PDB ID 8QN8, *Msm* us-fork promoter–HelD–σ$^A$–RNAP complex State II: EMD-18656, PDB ID 8QU6, *Msm* us-fork promoter–HelD–σ$^A$–RNAP complex State III: EMD-18873, PDB ID 8R3M, *Msm* us-fork promoter–HelD$_{N-term}$–σ$^A$–RNAP complex State III: EMD-18851, PDB ID 8R2M, *Msm* HelD$_{N-term}$–σ$^A$–RNAP complex RP2-like: EMD-18956, PDB ID 8R6P, *Msm* σ$^A_{N-helix}$–RNAP complex RP2-like: EMD-18959, PDB ID 8R6R, *Msm* RNAP open complex: EMD-18650, PDB ID 8QTI, The mass spectrometry proteomics data are available in the ProteomeXchange Consortium via the PRIDE partner repository with the dataset identifier PXD046632. The data can be accessed at http://www.ebi.ac.uk/pride/archive/projects/PXD046632. PDB accession codes of additional structures used in this study: 7PP4, 5TW1, 5VI5, 6EE8, 4XLS, 6YXU, 6YYS, 5VI8, 5UHC Source data are provided with this paper.

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

## Acknowledgements

We thank the ESRF (especially Michael Hons), IBS and EMBL for access to the ESRF Krios beamline CM01; the CEITEC (especially Jiri Novacek) and the Czech Infrastructure for Integrative Structural Biology (CIISB), part of Instruct-ERIC, for access to the CEITEC Krios microscope and to the CMS facilities at BIOCEV (projects LM2018127 and LM2023042 by MEYS, and the European Regional Development Fund UP CIISB, CZ.02.1.01/0.0/0.0/18_046/0015974). We thank the EMCF IMG for access to instrumentation (supported by LM2023050 by MEYS, and by CZ.02.1.01/0.0/0.0/18_046/0016045 and CZ.02.01.01/00/23_015/0008205 by ERDF). This work was supported by Ministry of Education, Youth and Sports of the Czech Republic grant RNA for therapy (CZ.02.01.01/00/22_008/0004575) (J.W.), Czech Science Foundation Grant 23-06295S (J. Do, L.K., T.K., H.S., B.B.), Grant Agency of Charles University in Prague; GA UK 236823 (N.B.), Czech Academy of Sciences Grant 86652036 (J. Do), ELIXIR CZ research infrastructure project MYES LM2023055 (M.S.), and the project National Institute of virology and bacteriology (Program EXCELES, ID Project No. LX22NPO5103)—Funded by the European Union—Next Generation EU (N.B., P.S., L.K.).

## Author contributions

J. Do, L.K., and T. Kou conceived and supervised the project. T. Kov and T. Kou expressed and purified proteins for cryo-EM, T. Kou prepared cryo-EM grids, collected cryo-EM data, performed image processing and 3D reconstruction, and built initial models together with T. Kov, T. Kov, N.B., H.S., M.T., V.V.H., and J. Du did cloning, protein purifications and IPs. A.K., M.H., and N.B. performed the mass spectrometry and protein analysis. N.B., P.S., and M.S. performed and analyzed the HelD binding assays. BB performed the transcription experiments. M.T. and K.A. performed NTP hydrolysis experiments. T. Kov and J. Do refined atomic models. T. Kov, J. Do, L.K., and T. Kou wrote the paper and together with H.Š., B.B., N.B., M.S., and J.W. prepared figures. M.S. performed statistical analyses. All authors discussed the paper and contributed to the interpretation.

## Competing interests

The authors declare no competing interests.
