## [Peer Review File · Nature Communications]

REVIEWER COMMENTS

Reviewer #1 (Remarks to the Author):

The authors report many new structures which add up to suggest important features of HelD/R function. The paper also includes functional studies with respect to the coupling between the ATPase activity of HelD, transcription initiation, and Rifampicin binding. Taken together, there is a host of new information that will be of interest to a wide range of researchers.

I do have a number of questions/points that I would ask the authors to consider prior to publication:

- (1) In terms of total transcription (not relative to -RIF) 20% down + HelD would seem to counteract the 20% improvement seen at low RIF concentrations. One should compare the actual total amount of transcription instead of normalizing relative to -RIF here if one wants to support the hypothesis that the presence of HelD protects against RIF by generating more transcript. Based on my current reading, the effects might cancel out.
- (2) The structures provide an amazing set of points in which to consider possible functions of HelD throughout the transcription cycle. However, as there is no time arrow in this method, one can not strictly assemble the structures in an order. This would contrast with new, time-resolved cryoEM methods (which may be an obvious direction for future work!). This should be acknowledged, perhaps in the discussion as a caveat for the model.
- (3) Whether HelD remains bound from termination through re-initiation will depend on the relative rates of rebinding a promoter and the ATP-stimulated rate of HelD dissociation. It occurs to me that given this, perhaps under low energy conditions (low ATP), HelD would stay bound, but under exponential growth (high ATP), it seems doubtful.
- (4) Perhaps I have missed it, but can the authors clarify what the product of the stalled-RNAP release reaction is? Does HelD release the polymerase solely through binding energy or is ATP-hydrolysis required. In either case, is the result a free HelD-RNAP complex or does HelD also dissociate? This will be critical for judging the possibility of a single HelD-RNAP complex remaining bound from rescue through re-initiation.
- (5) Relatedly, the stimulation of transcription by HelD has previously been reported by the same group. Interestingly, at moderate amounts of HelD a two-fold increase was observed. However, at higher amounts of HelD, inhibition was observed (<https://www.ncbi.nlm.nih.gov/pmc/articles/PMC4005671/>, Figure 4A). If HelD were to rescue stalled polymerases and then remain bound up through initiation, I wouldn't expect this kind of dependence. These data rather suggest that a concentration-dependent rebinding of HelD begins to limit transcription.
- (6) In the multi-round experiments, why are protein and DNA pre-incubated prior to the addition of RIF and NTPs. This would seem to allow for a burst of activity before RIF binding (i.e., the complexes that get past 2-3 nt will no longer bind RIF). Are the results the same if the reactions are initiated via the addition

of DNA.

(7) With respect to the alternative pathway to initiation where the authors propose a HelD containing initiation complex: I believe this model would predict a HelD-dependent stimulatory effect under single-round conditions in the presence of rifampicin. Since the proposed reaction all takes place without dissociating from the promoter, the presence of competitor under single-round conditions should not preclude one from observing the enhancement. Given the large number of results reported here, I am reticent to suggest more. That said, this experiment could be readily completed and would serve to support the interesting and exciting model proposed by the authors.

Reviewer #2 (Remarks to the Author):

Koval et al. present pull-down data suggesting that mycobacterial HelD, originally identified as an RNA polymerase (RNAP) recycling factor, can jointly bind RNAP with sigma factors and transcription factor RbpA, but not with transcription factor CarD, both in exponential and in stationary phase. Resorting to recombinantly produced proteins, they then assembled RNAP-sigmaA-HelD-RbpA holoenzymes and complexes with promoter DNA (with or without artificial transcription bubble and without or with initially transcribed RNA) and subjected the preparations to cryoEM-based structural analysis. They observed two states of HelD/RbpA-modified sigmaA-holoenzyme, in which HelD was bound in a similar fashion as previously observed for HelD-RNAP recycling complexes. Several states of HelD/RbpA-modified closed and open complexes in turn illustrate progressive expelling of HelD regions from the RNAP primary and secondary channel, respectively. Structural comparisons explain why binding of HelD and CarD is mutually exclusive. The authors also observed that RNAP-bound HelD exhibited enhanced ATPase activity. Using HelD variants deficient in ATP binding and/or hydrolysis, they showed that both ATP binding and hydrolysis contribute to HelD release, and that HelD release was not significantly influenced by sigmaA or RbpA. Consistent with the structural analysis, open promoter DNA, but not closed promoter DNA, also stimulated HelD release, and both types of DNA enhanced the ATP effect on HelD release. Also consistent with the structural data, HelD did not affect the affinity of RNAP for the initiating NTP, unlike some other secondary channel-binding factors. Finally, the authors showed that HelD has a protective effect against the transcription inhibitor rifampicin, as it distorts the rifampicin binding site on RNAP.

The data presented are novel and interesting, the work conducted appears technically sound. The results explain how HelD-mediated RNAP recycling is compatible with subsequent formation of holoenzymes that can initiate new rounds of transcription, how HelD is released by promoter DNA and ATP binding/hydrolysis, and how HelD can protect against the transcription inhibitor rifampicin. The findings should be of interest to a wide audience. This reviewer has only minor points to raise.

Specific comments:

1. The abbreviations used to denote the various structural states observed are very complicated, not really explained in the text and non-intuitive to this reviewer. It would be nice if the authors could find

simpler, more intuitive nomenclature, e.g. referring to the types of complexes visualized (i.e. holoenzymes, closed complexes, open complexes, initial transcription complexes).

2. Fig. 4c – colors of icons denoting proteins should be adjusted to the colors used in structure figures.
3. For all gel images in Figures/Supplementary Figures, types of gels and manner of staining should be indicated in the legends.
4. Fig. S1b, Fig. S17b – how were (relative) protein amounts quantified? Densitometry?
5. Fig. S19a – 11 primary data points are shown but 12 data points are plotted? Please indicate the GTP concentration for each lane in the top panel.
6. For angular distribution plots (particle projections), legends explaining color-coding are missing.
7. Particles seem to exhibit preferred orientations after vitrification, in some cases orientation preferences appear quite severe. Why did the preferred particle orientations not affect the cryoEM reconstructions? Alternatively, how did the preferred particle orientations affect the reconstructions?
8. The protective effect of HelD with respect to rifampicin appears to be small (Fig. 5). At the same time, HelD itself seems to have a dampening effect on transcription (Fig. S19). How can HelD exert a significant protective effect in vivo?
9. Source Data file is missing.
10. Authors should make structure coordinates and cryoEM reconstructions available to reviewers for inspection.

Reviewer #3 (Remarks to the Author):

Koval and Borah et al. present a study that structurally characterises the interplay between *Mycobacterium smegmatis* HelD and RNAP during the transcription cycle using a series of structures solved by cryo-EM. This manuscript will be of immense interest to the wider community, as it not only mechanistically outlines how HelD is released from RNAP, but also showcases the authors expertise in cryo-EM processing, and thus is suitable for publication in *Nature Communications* after a few minor points are addressed.

1. The final model of HelD release from RNAP is confusing and hard to interpret. I suggest the reviewers adopt a circular arrangement to emphasise each state. This would allow for more detail to be included, such as where ATP/GTP hydrolysis can stimulate the process. While the resolution of figure 6 may be reduced in the submitted version, the authors should ensure this is at the highest quality for revision (e.g. increase the line weight to make the components pop)

2. The authors should ensure the figures are arranged to match when they are first mentioned in the text, such as figure 4d and 4f. The section from line 424 to 441 could be rearranged to match current panels in figure 4 or visa versa.

3. A morph model of the sequential beta'-clamp closure would be useful to illustrate the progressive expelling of HeID

4. The PCh-loop is not annotated in Figure 2. This should be included for clarity.

5. Can the authors please comment on the FSC curve for the RP2- σ A N-helix complex (Supp Fig 8d). This should have a smooth return to 0 and suggests an issue with the dynamic mask used.

6. Can the authors please include source data for all SDS-PAGE and gels used throughout, such as figure 4. Is there a ladder to verify band shown is HeID? The contrast should be even for these across each panel

SUMMARY:

We thank all the reviewers for their insightful comments – they helped improve the manuscript! Based on the comments, we performed additional experiments, created new Figures and Movies, and modified the text accordingly. Also, we shortened (simplified) the title of the manuscript. For details, please see the responses below.

POINT-BY-POINT RESPONSE TO REVIEWERS:

Reviewer #1 (Remarks to the Author):

The authors report many new structures which add up to suggest important features of HelD/R function. The paper also includes functional studies with respect to the coupling between the ATPase activity of HelD, transcription initiation, and Rifampicin binding. Taken together, there is a host of new information that will be of interest to a wide range of researchers.

I do have a number of questions/points that I would ask the authors to consider prior to publication.

(1) In terms of total transcription (not relative to -RIF) 20% down + HelD would seem to counteract the 20% improvement seen at low RIF concentrations. One should compare the actual total amount of transcription instead of normalizing relative to -RIF here if one wants to support the hypothesis that the presence of HelD protects against RIF by generating more transcript. Based on my current reading, the effects might cancel out.

RESPONSE:

We appreciate this comment and agree that the dampening effect of HelD is an issue to consider while interpreting the *in vitro* transcription experiments.

HelD's role in rifampicin resistance is established. As described in the Hurst-Hess et al. paper (Figure 3a¹), HelD is one of the most highly induced genes in *Mycobacterium smegmatis* when exposed to a sub-lethal dose of rifampicin. That paper (Figure S3) also shows rifampicin sensitivity upon deletion of *HelD* (MSMEG_2174). Importantly, the paper by Surette et al.² shows that HelD was able to remove rifampicin from RNAP. However, *in vitro* transcription experiments in Hurst-Hess et al. (Figure S3³), similar to our experiments (Figure S18a), showed also a HelD dampening effect on transcription in their transcription system, indicating that it may be a general phenomenon in mycobacteria.

As suggested in comment #7, we also performed single-round transcription assays with and without HelD in the presence of rifampicin and obtained similar results as in multiple-round transcriptions presented in the original manuscript (Figure 5a).

To conclude, our experiments were able to replicate the published behavior of HelR and confirm the HelD's ability to alter transcription in response to rifampicin. However, the *in vitro* experiments might not be able to completely describe the complexity of the system, especially regarding the multi state process described in our model.

ACTION TAKEN:

We have addressed this issue in the manuscript. In Discussion, we mention both the general dampening effect of HelD on transcription as well as its effect on altering transcription inhibition by increasing concentrations of rifampicin. These two effects appear similar. We specify this also in the Figure 5a

legend (lines 481-483). Furthermore, to be clear about this issue, we acknowledge the limitations of the experimental set-up (lines 602-604,607-608).

(2) The structures provide an amazing set of points in which to consider possible functions of HelD throughout the transcription cycle. However, as there is no time arrow in this method, one cannot strictly assemble the structures in an order. This would contrast with new, time-resolved cryoEM methods (which may be an obvious direction for future work!). This should be acknowledged, perhaps in the discussion as a caveat for the model.

RESPONSE:

We thank the reviewer for the comment. Indeed, there is no strict event succession determined in the cryo-EM experiments we performed, and we have now acknowledged that in the model legend. On the other hand, we have tried to order the revealed structures into a comprehensive model according to the transcription initiation events already known in the field. In order to reveal the events tied to ATP binding/hydrolysis, we have tried one more time to visualize the ATP-bound form of the complex using the ATP hydrolysis non-capable variant of the HelD protein, however, failed to do so again. Probably, even the binding of ATP triggers some conformational changes in the complex which disfavor stable complex with ATP. Indeed, a time-resolved cryo-EM experiment might ultimately tackle this challenge. This is, however, out of the scope of this work.

ACTION TAKEN:

We have now placed the figures into a circular arrangement as suggested by reviewer 3. This is in an effort to provide a better interpretation of the whole model.

(3) Whether HelD remains bound from termination through re-initiation will depend on the relative rates of rebinding a promoter and the ATP-stimulated rate of HelD dissociation. It occurs to me that given this, perhaps under low energy conditions (low ATP), HelD would stay bound, but under exponential growth (high ATP), it seems doubtful.

RESPONSE:

Yes, it is a possibility that changing levels of ATP during cell growth phases might finetune HelD dissociation kinetics in a concentration dependent manner. However, as of now, we do not have an indication that changing ATP levels affect the overall HelD association with RNAP in the cell. In Figure S1a, about the same amount of HelD was bound to RNAP in both exponential and stationary phases. ATP concentration in *Msm* cells during exponential phase is around ~4 mM (Figure 3a⁴) and it is not dropping below 1 mM even during stress conditions (unpublished data of Dr. Knejzlik [IOCB, Prague] in line with other reports^{5,6}). To test the requirements of HelD for ATP for the release process, we varied the concentration of ATP. The results showed that the release was almost constant from 1 mM to 8 mM (see supplementary Figure S15). Furthermore, the release of HelD is stimulated not only when ATP is added but it is further stimulated when the RNAP complex interacts with promoter DNA, especially with the open complex. Hence, not all HelD is released by ATP alone, some is still retained by a fraction of RNAP molecules. We therefore believe the ATP concentration is not the triggering point of HelD release, rather a stimulatory factor as it is discussed in the manuscript.

ACTION TAKEN:

An additional experiment was performed (see above) and Figure S15 was added into the manuscript (Supplementary information), where it is mentioned in RESULTS (lines 378-380) and the issue is discussed in DISCUSSION (lines 574-578).

(4) Perhaps I have missed it, but can the authors clarify what the product of the stalled-RNAP release reaction is? Does HeLD release the polymerase solely through binding energy or is ATP-hydrolysis required. In either case, is the result a free HeLD-RNAP complex or does HeLD also dissociate? This will be critical for judging the possibility of a single HeLD-RNAP complex remaining bound from rescue through re-initiation.

RESPONSE:

The results in Figure 4 show a fraction of dissociated HeLD from the HeLD-RNAP complex. ATP binding alone aids this process, and hydrolysis further stimulates it (Figure 4e).

In the assay, RNAP is bound to the beads and incubated with HeLD, dissociated HeLD is collected from the supernatant. This means that only the fraction of HeLD dissociated from RNAP is quantified. In other words, this experiment does not detect HeLD-RNAP complexes.

Finally, with respect to the fraction of HeLD released by ATP, please, see our RESPONSE to your previous comment.

ACTION TAKEN:

We added a sentence to DISCUSSION clarifying this point (lines 578-583).

(5) Relatedly, the stimulation of transcription by HeLD has previously been reported by the same group. Interestingly, at moderate amounts of HeLD a two-fold increase was observed. However, at higher amounts of HeLD, inhibition was observed (<https://www.ncbi.nlm.nih.gov/pmc/articles/PMC4005671/>, Figure 4A). If HeLD were to rescue stalled polymerases and then remain bound up through initiation, I wouldn't expect this kind of dependence. These data rather suggest that a concentration-dependent rebinding of HeLD begins to limit transcription.

RESPONSE:

Thank you for the comment. To clarify, the mentioned previous results showing transcription stimulation were done with a different class of the HeLD protein, the class I from *Bacillus subtilis*. We have never observed a stimulatory effect of HeLD on transcription initiation in experiments with class II HeLD from mycobacteria and this seems to be the case also in a published work from other authors³. Class I HeLD differs from mycobacterial HeLD (class II) by some structural features and class I HeLD has been so far shown to bind only to the RNAP core. Class I HeLD recycles RNAP to make it available for new rounds of transcription. However, the class I HeLD release seems to be decoupled from transcription initiation^{7,8}. The exact step by step mechanism of RNAP recycling by Class I HeLD is not known at this point.

(6) In the multi-round experiments, why are protein and DNA pre-incubated prior to the addition of RIF and NTPs. This would seem to allow for a burst of activity before RIF binding (i.e., the complexes that get past 2-3 nt will no longer bind RIF). Are the results the same if the reactions are initiated via the addition of DNA.

RESPONSE:

We thank the reviewer for pointing out this unclarity. Before initiating the *in vitro* transcription with NTPs, all components of the transcriptional machinery (including RIF, when present) were mixed, and sufficient time (10 minutes) was allowed for equilibration. Therefore, there could not have been a burst of transcription before RIF binding.

ACTION TAKEN:

To clarify the order of addition of components in the *in vitro* transcription experiment, we changed the respective paragraph in the Materials and Methods section (lines 897-912).

(7) With respect to the alternative pathway to initiation where the authors propose a HelD containing initiation complex: I believe this model would predict a HelD-dependent stimulatory effect under single-round conditions in the presence of rifampicin. Since the proposed reaction all takes place without dissociating from the promoter, the presence of competitor under single-round conditions should not preclude one from observing the enhancement. Given the large number of results reported here, I am reticent to suggest more. That said, this experiment could be readily completed and would serve to support the interesting and exciting model proposed by the authors.

RESPONSE:

We appreciate this comment. Please, see our response to your comment 1 where we discuss single round conditions and the experiments we performed.

Reviewer #2 (Remarks to the Author):

Koval et al. present pull-down data suggesting that mycobacterial HelD, originally identified as an RNA polymerase (RNAP) recycling factor, can jointly bind RNAP with sigma factors and transcription factor RbpA, but not with transcription factor CarD, both in exponential and in stationary phase. Resorting to recombinantly produced proteins, they then assembled RNAP-sigmaA-HelD-RbpA holoenzymes and complexes with promoter DNA (with or without artificial transcription bubble and without or with initially transcribed RNA) and subjected the preparations to cryoEM-based structural analysis. They observed two states of HelD/RbpA-modified sigmaA-holoenzyme, in which HelD was bound in a similar fashion as previously observed for HelD-RNAP recycling complexes. Several states of HelD/RbpA-modified closed and open complexes in turn illustrate progressive expelling of HelD regions from the RNAP primary and secondary channel, respectively. Structural comparisons explain why binding of HelD and CarD is mutually exclusive. The authors also observed that RNAP-bound HelD exhibited enhanced ATPase activity. Using HelD variants deficient in ATP binding and/or hydrolysis, they showed that both ATP binding and hydrolysis contribute to HelD release, and that HelD release was not significantly influenced by sigmaA or RbpA. Consistent with the structural analysis, open promoter DNA, but not closed promoter DNA, also stimulated HelD release, and both types of DNA enhanced the ATP effect on HelD release. Also consistent with the structural data, HelD did not affect the affinity of RNAP for the initiating NTP, unlike some other secondary channel-binding factors. Finally, the authors showed that HelD has a protective effect against the transcription inhibitor rifampicin, as it distorts the rifampicin binding site on RNAP.

The data presented are novel and interesting, the work conducted appears technically sound. The results explain how HelD-mediated RNAP recycling is compatible with subsequent formation of holoenzymes that can initiate new rounds of transcription, how HelD is released by promoter DNA and ATP binding/hydrolysis, and how HelD can protect against the transcription inhibitor rifampicin. The findings should be of interest to a wide audience. This reviewer has only minor points to raise.

1) The abbreviations used to denote the various structural states observed are very complicated, not really explained in the text and non-intuitive to this reviewer. It would be nice if the authors could find simpler, more intuitive nomenclature, e.g. referring to the types of complexes visualized (i.e. holoenzymes, closed complexes, open complexes, initial transcription complexes).

RESPONSE:

Thank you for the comment. We indeed understand the set of abbreviations used to denote the various structural states are not simple. We have tried several variants of the abbreviations. We chose this variant of the abbreviations to be consistent with our previous work, that defined State I, II and III of the HeID-RNAP complex, and we refer to these states in the current work.

ACTION TAKEN:

We have now updated the abbreviations to contain type of complex (holoenzyme:holo, closed complex: RPc and transcription initiation promoter unwinding intermediate: RP2⁹). We included the most defining protein component and conformational state (State I, II and III).

Alternative naming:

State I HeID-holoenzyme: HeID-holo-I

State II HeID-holoenzyme: HeID-holo-II

State II upstream fork promoter HeID closed complex: us-fork-HeID-RPc-II

State III upstream fork promoter HeID closed complex: us-fork-HeID-RPc-III

State III upstream fork promoter HeIDN-term closed complex: us-fork-HeID_{N-term}-RPc-III

HeIDN-term transcription initiation promoter unwinding intermediate RP2 complex: HeID_{N-term}-RP2

σ^A -helix transcription initiation promoter unwinding intermediate RP2 complex: $\sigma^A_{N-helix}$ -RP2

2. Fig. 4c – colors of icons denoting proteins should be adjusted to the colors used in structure figures.

RESPONSE:

We thank the reviewer to pointing out this inconsistency.

ACTION TAKEN:

The colors of the proteins in Figure 4c were changed to match the structural data.

3. For all gel images in Figures/Supplementary Figures, types of gels and manner of staining should be indicated in the legends.

RESPONSE:

Thanks to the reviewer for this comment.

ACTION TAKEN:

We updated the figure legend texts wherever necessary to specify the gel type and staining used.

4. Fig. S1b, Fig. S17b – how were (relative) protein amounts quantified? Densitometry?

RESPONSE:

The reviewer is correct. Relative amounts of proteins were quantified by densitometry.

ACTION TAKEN:

We specified in the legend of Fig. S1b, Fig. S17b how protein amounts had been quantified.

5. Fig. S19a – 11 primary data points are shown but 12 data points are plotted? Please indicate the GTP concentration for each lane in the top panel.

RESPONSE:

We thank the reviewer for this comment. The [GTP] = 0 point was not shown in the Figure as transcription was not even performed in the absence of GTP (there would be no transcription without GTP).

ACTION TAKEN:

We have modified the text of figure S19a (now S20a) legend for clarification (Page S25, Paragraph 1)

6. For angular distribution plots (particle projections), legends explaining color-coding are missing.

RESPONSE:

We thank the reviewer to pointing out a missing information.

ACTION TAKEN:

We added the color-coding legend and description of the legend into the figure descriptions. In detail, every point on globe like plane is a particle orientation and the color scale represents the normalized density of views around this point. The scale is arbitrarily selected to illustrate differences in orientation distribution, the color scale runs from 0 (low, blue) to 0.0005 (high, red). We also mention the cryoEF program for angular distribution plotting in the method section.

7. Particles seem to exhibit preferred orientations after vitrification, in some cases orientation preferences appear quite severe. Why did the preferred particle orientations not affect the cryoEM reconstructions? Alternatively, how did the preferred particle orientations affect the reconstructions?

RESPONSE:

Indeed, some of the cryo-EM reconstructions exhibit some degree of orientation preference.

ACTION TAKEN:

We calculated the 3D FSC¹⁰ curves and the sphericity of the all reconstructions and added those to the cryo-EM analysis figures and Table S4. The RP2- σ^{AN} -helix 3D reconstruction has sphericity 0.823 and we can observe that the reconstruction is indeed slightly anisotropic. Nevertheless, the reconstruction is of a sufficient quality for interpretation.

8. The protective effect of HeLD with respect to rifampicin appears to be small (Fig. 5). At the same time, HeLD itself seems to have a dampening effect on transcription (Fig. S19). How can HeLD exert a significant protective effect in vivo?

RESPONSE:

We appreciate this comment. For our RESPONSE and ACTION TAKEN, please see the response to comment 1 of Reviewer #1

9. Source Data file is missing.

RESPONSE:

The original submission did not contain the Source Data file.

ACTION TAKEN

We now provide the Source Data file (Koval_et_al_Source Data_File.xlsx).

10. Authors should make structure coordinates and cryoEM reconstructions available to reviewers for inspection.

RESPONSE:

We appreciate the initiative for structure coordinates and 3D reconstructions inspection.

ACTION TAKEN

We provide the coordinates and respective cryo-EM reconstructions in a zip file (Koval_et_al_cryoEM_data.zip of size ~820MB) available at this link:

<https://cloud.uochb.cas.cz/owncloud/index.php/s/g9mHDq4C9A8Oh3l> password: Koval2024

We included the LocScale or LAFTER filtered mrc maps for better interpretation. These data are strictly confidential, please treat them as such.

Reviewer #3 (Remarks to the Author):

Koval and Borah et al. present a study that structurally characterizes the interplay between Mycobacterium smegmatis HelD and RNAP during the transcription cycle using a series of structures solved by cryo-EM. This manuscript will be of immense interest to the wider community, as it not only mechanistically outlines how HelD is released from RNAP, but also showcases the authors expertise in cryo-EM processing, and thus is suitable for publication in Nature Communications after a few minor points are addressed.

1. The final model of HelD release from RNAP is confusing and hard to interpret. I suggest the reviewers adopt a circular arrangement to emphasise each state. This would allow for more detail to be included, such as where ATP/GTP hydrolysis can stimulate the process. While the resolution of figure 6 may be reduced in the submitted version, the authors should ensure this is at the highest quality for revision (e.g. increase the line weight to make the components pop)

RESPONSE:

We appreciate the comment. We have updated the model figure into circular arrangement. However, we must note, that the cryo-EM structure experiments do not strictly determine the event succession as also noted by Reviewer #1 (comment 2). We acknowledged this limitation in the model legend. We also updated the naming of the individual states for more intuitive nomenclature. Our experiments do not exactly reveal at which stage ATP/GTP hydrolysis stimulates HelD release. It can potentially be any transition from panels e to f. Therefore we did not include it in the final model.

ACTION TAKEN:

We attach a high-resolution version of the model figure (Koval_et_al_Figure_6.png) and also all other main figures. We tested its readability in printed version on standard page format. In respect to the size of the Figure 6, we suggest a whole page should be dedicated for it to ensure readability in the printed version.

2. The authors should ensure the figures are arranged to match when they are first mentioned in the text, such as figure 4d and 4f. The section from line 424 to 441 could be rearranged to match current panels in figure 4 or vice versa.

RESPONSE:

We thank the reviewer for this suggestion.

ACTION TAKEN:

Figure 4 was rearranged to align with the text flow. Moreover, we added Mw ladders to the primary data strips.

3. A morph model of the sequential beta'-clamp closure would be useful to illustrate the progressive expelling of HeID.

RESPONSE:

We thank the reviewer for this suggestion, which will improve the interpretation of our results.

ACTION TAKEN:

We included a morph model (Koval_et_al_movie_4.mp4) of the observed structures in sequential order according to the proposed model in Figure 6. However, although we believe the order of the structures interpret the order of events the best, the true order of events is uncertain and we might be missing intermediates. We have acknowledged that in the model legend.

4. The PCh-loop is not annotated in Figure 2. This should be included for clarity.

RESPONSE:

We thank the reviewer for pointing out this unclarity.

ACTION TAKEN:

Done

5. Can the authors please comment on the FSC curve for the RP2-σA N-helix complex (Supp Fig 8d). This should have a smooth return to 0 and suggests an issue with the dynamic mask used.

RESPONSE:

We thank the reviewer for the comment. To clarify, we calculated the mentioned cryo-EM reconstruction with a very tight static mask, to partially mitigate a preferred orientation problem of this particular reconstruction.

ACTION TAKEN:

We calculated the 3D FSC¹⁰ curves and the sphericity of the all reconstructions and added those to the cryo-EM analysis figures and Table S4. The RP2-σAN-helix 3D reconstruction has sphericity 0.823 and we can observe that the reconstruction is indeed slightly anisotropic. Nevertheless, the reconstruction is of a sufficient quality for interpretation. The dip in the FSC curve is caused by reduced information in one spatial direction.

6. Can the authors please include source data for all SDS-PAGE and gels used throughout, such as figure 4. Is there a ladder to verify band shown is HeID? The contrast should be even for these across each panel.

RESPONSE:

Yes. As for the contrast, contrast settings within individual panels (for all the lanes) were the same.

ACTION TAKEN:

Figure 4g (4e in the original version) was rearranged and modified (see your comment 2 and our RESPONSE and ACTION TAKEN). The Mw marker is shown and the sizes of relevant bands (in kDa) are indicated. Moreover, we added into Mat&Met information about the Mw markers used in the study (name, manufacturer). As the system contained purified components, the band of this size could be only HeID. Moreover, gels used for primary data shown in the Figures are shown in the Source Data file.

References:

- 1 Hurst-Hess, K. *et al.* Mycobacterial SigA and SigB Cotranscribe Essential Housekeeping Genes during Exponential Growth. *mBio* **10** (2019). <https://doi.org:10.1128/mBio.00273-19>
- 2 Surette, M. D., Waglechner, N., Koteva, K. & Wright, G. D. HelR is a helicase-like protein that protects RNA polymerase from rifamycin antibiotics. *Molecular cell* **82**, 3151-3165 e3159 (2022). <https://doi.org:10.1016/j.molcel.2022.06.019>
- 3 Hurst-Hess, K. R., Saxena, A., Rudra, P., Yang, Y. & Ghosh, P. Mycobacterium abscessus HelR interacts with RNA polymerase to confer intrinsic rifamycin resistance. *Molecular cell* **82**, 3166-3177 e3165 (2022). <https://doi.org:10.1016/j.molcel.2022.06.034>
- 4 Knejzlik, Z. *et al.* The mycobacterial guaB1 gene encodes a guanosine 5'-monophosphate reductase with a cystathionine-beta-synthase domain. *FEBS J* **289**, 5571-5598 (2022). <https://doi.org:10.1111/febs.16448>
- 5 Tran, Q. H. & Unden, G. Changes in the proton potential and the cellular energetics of Escherichia coli during growth by aerobic and anaerobic respiration or by fermentation. *Eur J Biochem* **251**, 538-543 (1998). <https://doi.org:10.1046/j.1432-1327.1998.2510538.x>
- 6 Gengenbacher, M., Rao, S. P. S., Pethe, K. & Dick, T. Nutrient-starved, non-replicating Mycobacterium tuberculosis requires respiration, ATP synthase and isocitrate lyase for maintenance of ATP homeostasis and viability. *Microbiology (Reading)* **156**, 81-87 (2010). <https://doi.org:10.1099/mic.0.033084-0>
- 7 Pei, H. H. *et al.* The delta subunit and NTPase HeID institute a two-pronged mechanism for RNA polymerase recycling. *Nat Commun* **11**, 6418 (2020). <https://doi.org:10.1038/s41467-020-20159-3>
- 8 Newing, T. P. *et al.* Molecular basis for RNA polymerase-dependent transcription complex recycling by the helicase-like motor protein HeID. *Nat Commun* **11**, 6420 (2020). <https://doi.org:10.1038/s41467-020-20157-5>
- 9 Boyaci, H., Chen, J., Jansen, R., Darst, S. A. & Campbell, E. A. Structures of an RNA polymerase promoter melting intermediate elucidate DNA unwinding. *Nature* **565**, 382-385 (2019). <https://doi.org:10.1038/s41586-018-0840-5>
- 10 Tan, Y. Z. *et al.* Addressing preferred specimen orientation in single-particle cryo-EM through tilting. *Nature methods* **14**, 793-796 (2017). <https://doi.org:10.1038/nmeth.4347>

REVIEWERS' COMMENTS

Reviewer #1 (Remarks to the Author):

I would thank the authors for their thoughtful responses to my comments. In my opinion, the manuscript should be published. I have two final thoughts/recommendations:

(1) The title suggests an "alternative" pathway for initiation which seems a bit of an exaggeration. Initiation is regulated by many factors that interact with the polymerase on and off of the DNA. While the kinetics and structures may differ, I would favor the view where the overall MECHANISM of initiation is the same. Perhaps HelD couples termination to initiation or chaperones RNAP to prevent Rif binding, but I would not agree that it changes the mechanism of initiation.

(2) I recommend that the authors quantify the HelD dissociation experiments in terms of percent release. A control lane quantifying the amount of HelD bound to the beads to begin with can be used in this normalization and would make the results of these results easier to describe and interpret.

Reviewer #2 (Remarks to the Author):

In revising their manuscript, the authors have adequately addressed all comments raised by this reviewer. Congratulations on a very nice piece of research.

Reviewer #3 (Remarks to the Author):

I thank the authors for addressing all initial comments, with the notable inclusion of the circular schematic to illustrate the proposed model of transcription initiation. The overall comments have improve the manuscript and ensured the authors can demonstrate the significance of all their new cryo-EM structures. The manuscript by Koval and Borah et al. is now suitable for publication in *Nature Communications*

REVIEWERS' COMMENTS

Reviewer #1 (Remarks to the Author):

I would thank the authors for their thoughtful responses to my comments. In my opinion, the manuscript should be published. I have two final thoughts/recommendations:

(1) The title suggests an "alternative" pathway for initiation which seems a bit of an exaggeration. Initiation is regulated by many factors that interact with the polymerase on and off of the DNA. While the kinetics and structures may differ, I would favor the view where the overall MECHANISM of initiation is the same. Perhaps HelD couples termination to initiation or chaperones RNAP to prevent Rif binding, but I would not agree that it changes the mechanism of initiation.

RESPONSE:

Thank you for bringing up this point. We agree with the reviewer.

ACTION TAKEN:

We have changed the previous title to "Mycobacterial HelD connects RNA polymerase recycling with transcription initiation"

(2) I recommend that the authors quantify the HelD dissociation experiments in terms of percent release. A control lane quantifying the amount of HelD bound to the beads to begin with can be used in this normalization and would make the results of these results easier to describe and interpret.

RESPONSE:

Thank you for this comment. However, in our experimental setup, HelD dissociation was normalized relative to the amount of HelD released in the presence of ATP (set as 1). As for what remained on the beads, the signal was in some cases not in the range of our quantitative densitometric analysis (Supplementary Figure S14) and, therefore, not reliable to quantitate. So, to do this quantitation, we would have to repeat a number of experiments, and load lower amounts of protein onto the gel. To conclude, we believe that the quantitation that is used still reflects the relative efficiencies of HelD release depending on the presence/absence of additional factors (NTP, DNA).

Reviewer #2 (Remarks to the Author):

In revising their manuscript, the authors have adequately addressed all comments raised by this reviewer. Congratulations on a very nice piece of research.

RESPONSE:

Thank you.

Reviewer #3 (Remarks to the Author):

I thank the authors for addressing all initial comments, with the notable inclusion of the circular schematic to illustrate the proposed model of transcription initiation. The overall comments have improve the manuscript and ensured the authors can demonstrate the significance of all their new cryo-EM structures. The manuscript by Koval and Borah et al. is now suitable for publication in .
Nature Communications

RESPONSE:

Thank you.